# Synthesis of Tetrasubstituted Phosphorus Analogs of Aspartic Acid as Antiproliferative Agents

**DOI:** 10.3390/molecules27228024

**Published:** 2022-11-18

**Authors:** Xabier del Corte, Aitor Maestro, Adrián López-Francés, Francisco Palacios, Javier Vicario

**Affiliations:** Departamento de Química Orgánica I, Centro de Investigación y Estudios Avanzados “Lucio Lascaray”, Facultad de Farmacia, University of the Basque Country, UPV/EHU Paseo de la Universidad 7, 01006 Vitoria-Gasteiz, Spain

**Keywords:** Reformatsky reaction, tetrasubstituted α-aminophosphonates, aspartic acid, antiproliferative effect

## Abstract

An efficient general method for the synthesis of a wide family of α-aminophosphonate analogs of aspartic acid bearing tetrasubstituted carbons is reported through an aza-Reformatsky reaction of α-iminophosphonates, generated from α-aminophosphonates, in an umpolung process. In addition, the α-aminophosphonate substrates showed in vitro cytotoxicity, inhibiting the growth of carcinoma human tumor cell lines A549 (carcinomic human alveolar basal epithelial cell) and SKOV3 (human ovarian carcinoma). In view of the possibilities in the diversity of the substituents that offer the synthetic methodology, an extensive profile structure–activity is presented, measuring IC_50_ values up to 0.34 µM in the A549 and 9.8 µM in SKOV3 cell lines.

## 1. Introduction

The increase in life expectancy is one of the greatest achievements of humankind, which is also linked to far-reaching consequences with implications for nearly all socioeconomic sectors [1]. According to a study published in The Lancet in 2018 [2], by 2040, it is expected that 59 countries will have an average life expectancy of more than 80 years. In particular, the average life expectancy in Spain is predicted to be the highest in the world and will reach 85.8 years. In the past, the main roots of mortality were associated to infectious and parasitic diseases but, due to the phenomenon of population ageing, chronic and degenerative diseases have become the main concern of all healthcare systems worldwide. Accordingly, cancers figure among the leading causes of morbidity and mortality worldwide and have become one of the world’s largest health problems [3].

The systemic treatment of cancer implies a combination of surgery, chemotherapy and radiation. Other options include immunotherapy, targeted therapy, laser or hormonal therapy [4]. Chemotherapeutic agents possess the ability to travel throughout the body, and selectively destroy fast-growing malignant cells [5]. Here, in the first front of battle, drug discovery plays a crucial role in this area through the synthesis and characterization of drug candidates and the evaluation of their anticancer properties, prior to the subsequent clinical trials. Despite the strong efforts made in the last decades for the development of efficient chemotherapeutic agents, the continuous search for newer, safer and more potent cytotoxic drugs is an essential task in science, especially due to the known ability of cancer cells to develop resistance to the known therapies [6,7].

Among the innumerable amount of potentially chemotherapeutic molecules, we focused, in this case, on organophosphorus compounds. In particular, phosphonic acids and their esters are a family of compounds, characterized by the presence of a stable C-P bond in their structure, that show interesting and useful biological properties [8], including anticancer activity, such as the case of cyclophosphamide [9] or zolendronate [10,11]. Specifically, α-aminophosphonic acids are bioisosters of α-amino acids, where the flat carboxylic acid group has been replaced by a tetrahedral phosphonic acid group that shows the fully oxidized phosphorus atom at the core [12]. Due to this isosteric substitution, α-aminophosphonic acid scaffold is able to mimic the tetrahedral geometry and negative charge development found in the transition state of peptide cleavage, thus inhibiting enzymes implied in proteolysis processes (Figure 1) [13,14,15]. Considering this, it is easy to anticipate that a number of α-aminophosphonic acid derivatives show interesting biological activities, such as herbicidal [16,17], antimicrobial [18,19,20,21] or antioxidant [22,23], and some of them have been reported as potential drugs, in particular, for the treatment of infectious diseases [24,25]. Remarkably, some α-aminophosphonate derivatives have been described as anticancer [26,27,28,29] agents.

Aspartic acid is one of the 20 building block α-amino acids of proteins that is known to have pharmacological activity at some glutamate receptors [30]. The most interesting feature of the structure of aspartic acid is the presence of a second acidic side chain that may interact with other amino acids, enzymes or proteins in the body. In the context of this research, following the biosisosterism approach, two possible phosphorated analogs may be designed from aspartic acid scaffold **I**: β-phosphorylated α-aminoacids **II** or β-carboxylic α-aminophosphonates **III** (Figure 1).

While the synthesis of phosphonate analogs of aspartic acid by the isosteric substitution of the β-carboxylic group is well documented [31,32,33], the preparation of the parent α-aminophosphonate analogs **III** by the substitution of the α-carboxylic moiety has received less attention, and most of the substrates are reported as single examples of general methods leading to α-aminophosphonates or aspartic acid derivatives [32,34,35]. In particular, there are a few examples reported for the synthesis of tetrasubstituted α-aminophosphonates derived from aspartic acid as concrete examples of the scope of reactions that imply C-C or C-P bond formation [36,37,38,39]. It is well known that the development of reactions leading to the formation of tetrasubstituted carbons is a challenging task, due to the lack of reactivity of the substrates, derived from the generation of a highly crowded structure.

In this context, the aza-Reformatsky reaction is a widely used method for the synthesis of β-amino acids [40,41,42,43], as well as for the synthesis of biologically active molecules [44,45,46]. During the last years, the use of dialkylzinc reagents has emerged as an alternative to Zn dust [47,48,49,50]. In this regard, very recently, we have reported an enantioselective aza-Reformatsky reaction of α-phosphorylated ketimines that leads to the formation of tetrasubstituted α-aminophosphonate derivatives [51]. In view of the interesting properties of aspartic acid derivatives and the potential of the isosteric substitution of a carboxylate by a phosphonate group, the corresponding phosphorus analogs of aspartic acid may be very interesting substrates from a biological point of view. According to this, as part of our ongoing research into the identification of new chemotherapeutic agents [52,53,54], with a special focus on organophosphorus derivatives [55,56,57], we thought that the preparation of a wide family of phosphorated analogs of aspartic acid and the study of their anticancer properties would be an interesting contribution to the field of organic and medicinal chemistry. For all the reasons mentioned above, herein, we report a general method for the synthesis of tetrasubstituted phosphorated analogs of aspartic acid through an aza-Reformatsky reaction of α-ketiminophosphonates and the evaluation of their in vitro cytotoxic activity against several cancer cell lines.

## 2. Results and Discussion

### 2.1. Chemistry

During the last years, we have been involved in the synthesis of α-aminophosphonate derivatives through the addition of carbon nucleophiles to imines. The most remarkable feature of our approach is that α-iminophosphonate electrophiles are generated by the oxidation of the parent α-aminophosphonates. Thus, the global reaction can be considered as an umpolung process, where the nucleophilic character of α-aminophosphonate species has been inverted. In particular, following this approach, we have reported the enantioselective synthesis of indolyl phosphoglycines [58] by the addition of indole derivatives to α-phosphorated aldimines and the nucleophilic addition of cyanide [59], organometallics [60] or nitromethane species [61] to α-phosphorated ketimines, for the preparation of diverse tetrasubstituted α-aminophosphonate derivatives. More recently, we have extended this strategy to the enantioselective Reformatsky reaction, using α-ketiminophosphonates as the electrophile source [51].

Following this last approach, we tested the aza-Reformatsky reaction of different imines **1–4** with ethyl iodoacetate, under dry air atmosphere, in the presence of dimethylzinc, affording the corresponding β-aminoesters **4–6** very good yields when aldimines or activated ketimines were used (Figure 2).

Initially, the simple *N*-tosylimine **1** derived from benzaldehyde (R = H) was proved to be an excellent substrate for the reaction, affording β-phenylalanine derivative **5** a very good yield (Figure 2). Next, we were intrigued whether the reaction could also be applicable to α-iminoesters and, for this reason, we tried the same reaction conditions using *N*-tosyl-protected α-iminoester **2** (R = CO_2_Et) as a substrate. In this case, the reaction also proceeded efficiently to afford quaternary aspartic acid derivative **6** (Figure 2). In a similar way, using benzaldehyde-derived α-iminophosphonate **3a** (R = P(O)(OMe)_2_), the tetrasubstituted aspartic acid derivative **7a** was obtained in an excellent yield (Figure 2). However, the reaction conditions, including the required dimethylzinc and the reaction time, presented high variations depending on the structure of the imine. On the other hand, we also tested a non-activated ketimine **4**, derived from acetophenone. Unfortunately, in this case, the Reformatsky product **8** was not observed, and the starting materials were recovered unaltered. 

The substrates obtained from the aza-Reformatsky reaction were fully characterized on the basis of their ^1^H, ^31^P, and ^13^C NMR, IR spectra and HRMS (see Appendix A). The most characteristic pattern for these compounds in the ^1^H NMR spectrum is the signals corresponding to the two protons of the methylene group next to the tetrasubstituted carbon at δ~3.5 ppm, which, because of the presence of a chiral center in the structure, have a diastereotopic character and appear as two independent signals. In the particular case of phosphorated aspartic acid derivative **7a**, those signals appear as two double doublets at δ = 3.59 and 3.46 ppm, showing a reciprocal geminal coupling of ^2^*J*_HH_ = 16.4 Hz and additional vicinal couplings with the phosphorus atom of ^3^*J*_PH_ = 22.7 and 10.7 Hz, respectively. Accordingly, the dimethyl phosphonate moiety in **7a** is seen as two representative intense doublets at δ_H_ = 3.46 ppm (^3^*J*_PH_ = 10.7 Hz) and δ_H_ = 3.48 ppm (^3^*J*_PH_ = 10.5 Hz), typical for the diastereotopic methoxy groups at the phosphonate moiety. Remarkably, the signal corresponding to the NH group of **6a** appears as a thin doublet at δ_H_ = 6.17 ppm, that slowly interchanges with D_2_O, showing a strong coupling with the phosphorus atom of ^3^*J*_PH_ = 11.2 Hz, which may be attributable to a weak acidic character of the sulfonamide moiety.

Likewise, in the ^13^C NMR spectrum of phosphorylated derivative **7a**, undoubtedly, the most characteristic signal is the doublet corresponding to the chiral quaternary carbon (DEPT) at δ_C_ = 62.1 ppm, which shows a very strong *ipso* coupling with the phosphorus atom of ^1^*J*_PC_ = 153.8 Hz. The methylene group next to the chiral carbon appears as a doublet at δ_C_ = 54.0 ppm with a geminal coupling with the phosphorus atom of ^2^*J*_CP_ = 7.0 Hz, weaker than expected, possibly due to an unfavorable angle in terms of the coupling, which can be related to a distortion at the topology at the sp^3^ chiral carbon, attributed to the high steric hindrance present at the quaternary center. The presence of the ester group is evident from the chemical shift at δ_C_ = 170.2 ppm, typical for carboxylic groups, which appears as a doublet, coupled with the phosphorus atom with a vicinal coupling constant of ^3^*J*_PC_ = 8.0 Hz. The fact that the vicinal C-P coupling is stronger than the geminal supports the proposed distortion of the bonding angles at the quaternary carbon, as expected from the high steric crowding.

In congruity with the proposed structure, the Heteronuclear Multiple Bond Correlation Spectroscopy (HMBC) spectrum of **7a** presents clear correlations of both diastereotopic methylene protons with the carbonyl group, the chiral tetrasubstituted carbon and the quaternary aromatic carbon of the phenyl substituent.

Next, in view of the efficient protocol achieved for the aza-Reformatsky reaction, we focused our efforts on the extension of the reaction to the use of several α-ketiminophosphonate substrates **3**. In this regard, we considered the synthesis of α-iminophosphonate substrates **3** from a formal oxidation of tertiary aminophosphonates **9**. Then, the subsequent addition of a nucleophile species would afford tetrasubstituted aminophosphonates **11**. Therefore, this synthetic approach can be considered globally as a route for the generation of tetrasubstituted α-aminophosphonates by the substitution of hydrogen in a trisubstituted α-aminophosphonate by a nucleophilic reagent and the complementary process (‘umpolung reaction’) to the typical electrophilic substitution of trisubstituted α-amino-phosphonates **9** leading to functionalized α-aminophosphonates **10** (Figure 3).

Following this approach, α-ketiminophosphonates **3** were first generated by a formal oxidation of α-aminophosphonates **9**, following the procedure developed by our research group. Then, using dimethylphosphonate-substituted imines and ethyl iodoacetate, 19 tetrasubstituted aspartic acid analogs **7**, bearing different alpha-aromatic substituents, were efficiently synthesized (Figure 4).

In addition to the model reaction, using an imine with a simple phenyl substituent (Figure 4, **7a**), the reaction tolerates the presence of *para* and *meta* alkyl substituents at the aromatic ring (Figure 4, **7b–c**), as well as strong electron donating groups at the *para* position of the aromatic imine (Figure 4, **7d**). Several halogen-substituted aromatic ketimines were also successfully used in the reaction, including aromatic rings containing bromine (Figure 4, **7e**), chlorine (Figure 4, **7f–i**) or fluorine (Figure 4, **7j–n**) atoms at diverse positions and including a perfluorophenyl substituent (Figure 4, **7o**). Furthermore, an excellent result was observed using aromatic imines substituted with electron-withdrawing groups such as *p*-trifluoromethyl or *p*-nitro substituents (Figure 4, **7p–q**). The reaction can even be extended to the use of ketimines holding heteroaromatic or biphenyl substituents (Figure 4, **7r–s**).

Next, the synthetic procedure was extended to the use of other different alkyl iodoacetates. The reaction using methyl iodoacetate and a *p*-fluorophenyl substituted α-ketiminophosphonate affords the corresponding tetrasubstituted α-aminophosphonate **12** an excellent yield (Figure 5). Under the same conditions, the reaction using benzyl iodoacetate efficiently yields the benzyl-protected analogs of aspartic acid **13a–b** (Figure 5). 

In order to obtain the carboxylic acid and/or phosphonic acid derivatives of the aspartate analogs, the hydrolysis of the obtained substrates was attempted. However, under acidic or basic treatment, compounds **7a**, **12** or **13b** led to the formation of complex mixtures. On the contrary, the treatment of benzylester **13b** under hydrogen pressure in the presence of a palladium catalyst afforded the corresponding carboxylic acid **14** an almost quantitative yield (Figure 6).

While trying to further understand the nature of the reaction, some control experiments were performed using non-conventional haloacetate derivatives. For instance, the use of α-branched iodoacetate (Figure 7a) resulted in a complete loss of the reactivity that may be explained due to the electronic effect of the fluorine, which reduces the nucleophilicity of the intermediate species. In contrast, the use of a bulkier phosphonate instead of the ester group increases the steric demand of the nucleophile (Figure 7b). Finally, when using ethyl 3-iodopropionate, the formation of the enolate intermediate does not occur, and the corresponding organozinc halide species may be formed, which are not usual in nucleophilic additions due to their low reactivity (Figure 7c). 

Based on these control reactions, as well as on the reaction pathways proposed for similar processes [62,63], we theorize a tentative catalytic cycle in which a combination of electronic and steric effects and the stability of the zinc-enolate intermediate are considered (Figure 8). 

As has been addressed above, aspartic acid derivatives have proved to have assorted pharmacological activities [30]. Thus, in view of the described benefits of the isosteric substitution of a carboxylate by a phosphonate group [12], next, we explored the applications of the synthesized phosphorated analogs of aspartic acid as anticancer agents.

### 2.2. Biological Results

In vitro cytotoxicity of the phosphorated aspartic acid derivatives was evaluated by testing their antiproliferative activities against several human cancer cell lines. Cell counting kit (CCK-8) assay was used for the evaluation of growth inhibition. Moreover, non-malignant MRC5 lung fibroblasts were tested for studying selective toxicity [64] and chemotherapeutic doxorubicin is used as a reference value. In addition, trisubstituted aminophosphonate **9a** (Imine precursor, R^1^ = H, R^2^ = P(O)(OMe)_2_) [59] and tetrasubstituted aminophosphonate **18** (R^1^ = Me, R^2^ = P(O)(OMe)_2_) [60] are used as templates in order to evaluate the effect of the substituents (Table 1).

In a preliminary study, the most simple β-alanine substrate **5** showed some cytotoxicity against the A549 cell line, with modest IC_50_ values of 18.68 ± 2.16 µM but high selectivity if compared with the MCR5 cell line (Table 1, Entry 1). A slightly improved IC_50_ value of 14.17 ± 0.41 µM was observed for aspartic acid ester **6** in the same cell line and, remarkably, the isosteric substitution of the ester group by a phosphonate group in **7a** resulted in a significantly better IC_50_ value of 2.66 ± 0.26 µM, still with a high selectivity toward non-malignant cells (Table 1, Entries 2–3).

The presence of the ester group, however, proved to be very relevant to the antiproliferative activity of compound **7a** since tetrasubstituted α-aminophosphonate **18**, lacking such a substituent, showed no cytotoxicity against the A549 cell line and trisubstituted α-aminophosphonate **9a** showed a higher IC_50_ value of 17.56 ± 1.3 µM (Table 1, Entries 4–5). None of the tested compounds showed in vitro toxicity against the SKOV3 cell line.

This first test of our substrates demonstrated that the best results in activity are obtained when both α-phosphonate and β-carboxylate groups are present in the structure, forming a phosphorus analog of aspartic acid. For this reason, next, we performed a study of the effect of the substitution at the α-aromatic ring of the aspartic acid analogs into the cytotoxicity against the SKOV3 and A549 cell lines (Table 2).

The introduction of methyl groups into a bioactive structure results in a more lipophilic character, often resulting in an improved ability of molecules to cross cell membranes [65,66]. Indeed, para and meta tolyl-substituted aminophosphonates **7b** and **7c** presented improved IC_50_ values of 0.34 ± 0.04 and 2.00 ± 0.52 µM in the A549 cell line if compared with the model structure **7a** (Table 2, Entries 2–3 vs. Entry 1). Although compound **7c** showed a high selectivity if compared with the non-malignant cells, derivative **7b** showed some toxicity toward the MCR5 cell line. Unfortunately, no activity was observed against the SKOV3 cell line for both compounds (Table 2, Entries 2–3). In addition, *p*-trichloromethylthiophenyl derivative **7d** showed good IC_50_ values of 6.43 ± 0.64 and 4.41 ± 0.29 µM in the SKOV3 and A549 cell lines, respectively, but it presented a very low selectivity toward the MRC5 cell line (Table 2, Entry 4).

Next, the effect of the substitution of the α-aromatic ring with different halogen atoms was explored. First, *p*-bromo-substituted derivative **7e** proved to be a very good growth inhibitor of the A549 cell line with an IC_50_ value of 1.08 ± 0.09 µM, and a good selectivity if compared to the SKOV3 or MCR5 cell lines, which showed values over 50 µM (Table 2, Entry 5). *p*-Chlorophenyl derivative **7f** showed very good toxicity against the A549 cell line and some activity in the SKOV3 cell line with IC_50_ values of 3.09 ± 0.14 and 24.12 ± 1.45 µM, respectively, and a good selectivity against the healthy cells (Table 2, Entry 6). However, the *m*-chlorophenyl isomer **7g** was found to be less effective than the para isomer and toxic against the healthy cells (Table 2, Entry 7). Remarkably, the combination of both substituents in substrate **7h** resulted in a strong cytotoxic effect against all the cell lines and a total lack of selectivity (Table 2, Entry 8). Likewise, the combination of a *meta*-chloro and a *para*-methoxy group in the aromatic ring of **7i** resulted in a very selective activity against the A549 cell line, with an IC_50_ value of 1.44 ± 0.15 µM (Table 2, Entry 9).

Although, generally, the effect of fluorine substituents on the activity of organic compounds is rather difficult to predict, it is well known that the introduction of fluorine atoms into bioactive molecules very often leads to increased activities [67,68,69]. With this in mind, we explored the effect of the introduction of different fluorine-containing phosphorated analogs of aspartic acid **7j–p**. First, the introduction of a *para*-fluorine substituent in **7j** had a negative effect on the cytotoxicity if compared with the model phenyl-substituted compound **7a** (Table 2, Entry 10 vs. Entry 1). On the contrary, the *meta-* and *ortho*-substituted isomers **7k** and **7l**, compared with the model substrate **7a**, showed a very good cytotoxic effect and a high selectivity against the A549 cell line and improved IC_50_ values of 0.59 ± 0.09 and 0.90 ± 0.12 µM were observed, respectively (Table 2, Entries 11 and 12 vs. Entry 1). Those values could not be improved by the combination of two fluorine substituents at the aromatic ring and, although good IC_50_ values of 2.24 ± 0.31 and 5.70 ± 0.70 µM were obtained for difluoro-substituted substrates **7m–n** in the A549 cell line, those were significantly higher than the values obtained for mono-substituted substrates **7k-l** (Table 2, Entries 13 and 14 vs. Entries 11 and 12). Moreover, perfluorophenyl derivative **7o** was found to be active against the A549 and SKOV3 cell lines with IC_50_ values of 3.65 ± 0.21 and 20.46 ± 2.75 µM, respectively, with a good selectivity toward the healthy cells (Table 2, Entry 15).

Surprisingly, the introduction of a *para*-trifluoromethyl electron-withdrawing group at the aromatic ring had a positive effect on the cytotoxicity of substrate **7p** in the SKOV3 cell line, while it had a negative effect for the A549 cell line. In this case, IC_50_ values of 9.80 ± 0.60 and 20.3 ± 1.14 µM were obtained for each cell line (Table 2, Entry 16 vs. Entry 1) and no toxicity was observed in the MCR5 cells. Aspartic acid analog **7q**, bearing other electron poor aromatic substituents, such as a *p*-nitrophenyl group, showed very good toxicity in the A549 cell line and a high selectivity with an IC_50_ value of 0.67 ± 0.06 µM (Table 2, Entry 17). Finally, heteroaromatic or biphenyl α-substituted substrates **7r** and **7s** presented very good IC_50_ values of 1.04 ± 0.28 and 1.48 ± 0.40 µM in the A549 cell line and moderate cytotoxicity against the SKOV3 cell line, with IC_50_ values of 11.77 ± 0.60 and 17.01 ± 1.22 µM, respectively. Both compounds were found to be selective if compared with the non-malignant cells (Table 2, Entries 18 and 19).

In our next part of this research, we evaluated the influence of the ester group on the antiproliferative activity of our substrates. Considering this, first, we tested the in vitro cytotoxicity of methyl and benzyl ester derivatives **12** and **13a**. Both compounds showed higher IC_50_ values of 24.11 ± 4.01 and 12.75 ± 2.38 µM in the A549 cell line, respectively, if compared with their ethyl ester derivative **7k** and no toxicity against the SKOV3 cell line (Table 2, Entries 20 and 21 vs. Entry 11). The activity of a second benzyl ester derivative **13b** was also tested and that, as in the previous case, presented less toxicity than its ethyl ester derivative **7a** (Table 2, Entry 22 vs. Entry 1). Moreover, carboxylic acid derivative **14** was found also to present less activity than its ethyl ester derivative **7a** (Table 2, Entry 23 vs. Entry 1). Taking into account the results obtained in this experiment, we concluded that ethyl esters are far superior to other esters or carboxylic acid derivatives.

Finally, due to the presence of a stereogenic carbon in the structure of our phosphorated aspartic acid analogs **7**, one of the final questions to be addressed is which is the real activity of the two individual enantiomers. For that reason, we performed the enantioselective synthesis of α-aminophosphonate **7k**, using both enantipoure isomers of 9-antracenyl-substituted BINOL in the aza-Reformatsky reaction. Both enantiomers of **7k** were prepared in 99% ee and, then, two enantiopure samples were obtained by a subsequent crystallization. However, no significant differences were observed for the *R* or *S* enantiomers if compared with the racemic sample with IC_50_ values of 1.61 ± 0.20 and 1.56 ±0.39 µM in the A549 cell line, respectively (Figure 9).

## 3. Materials and Methods

### 3.1. Chemistry

#### 3.1.1. General Experimental Information

Solvents for extraction and chromatography were technical grade. All solvents used in reactions were freshly distilled from appropriate drying agents before use. All other reagents were recrystallized or distilled as necessary. All reactions were performed under an atmosphere of dry nitrogen. Analytical TLC was performed with silica gel 60 F_254_ plates. Visualization was accomplished by UV light. ^1^H, ^13^C, ^31^P and ^19^F-NMR spectra were recorded on a Varian Unity Plus (Varian Inc., NMR Systems, Palo Alto, CA, USA) (at 300 MHz, 75 MHz, 120 MHz and 282 MHz, respectively) and on a Bruker Avance 400 (Bruker BioSpin GmbH, Rheinstetten, Germany) (at 400 MHz for ^1^H, and 101 MHz for ^13^C). Chemical shifts (δ) are reported in ppm relative to residual CHCl_3_ (δ = 7.26 ppm for ^1^H and δ = 77.16 ppm for ^13^C NMR) and using phosphoric acid (50%) or HF as external reference (δ = 0.0 ppm) for ^31^P and ^19^F NMR spectra. Coupling constants (*J*) are reported in Hertz. Data for ^1^H NMR spectra are reported as follows: chemical shift, multiplicity, coupling constant, integration. Multiplicity abbreviations are as follows: s = singlet, d = doublet, t = triplet, q = quartet, m = multiplet. ^13^C NMR peak assignments were supported by distortionless enhanced polarization transfer (DEPT). High resolution mass spectra (HRMS) were obtained by positive-ion electrospray ionization (ESI). Data are reported in the form *m*/*z* (intensity relative to base = 100). Infrared spectra (IR) were taken in a Nicolet iS10 Thermo Scientific spectrometer (Thermo Scientific Inc., Waltham, MA, USA) as neat solids. Peaks are reported in cm^–1^.

#### 3.1.2. Compounds Purity Analysis

All synthesized compounds were analyzed by HPLC to determine their purity. The analyses were performed on an Agilent 1260 infinity HPLC system (Agilent, Santa Clara, CA, USA) using a CHIRALPAK^®^ IA column (5 μm, 0.54 cm ø × 25 cm, Daicel Chiral Technologies, Illkirch Cedex, France) at room temperature. All the tested compounds were dissolved in dichloromethane, and 5 μL of the sample was loaded onto the column. Ethanol and heptane were used as the mobile phase, and the flow rate was set at 1.0 mL/min. The maximal absorbance at the range of 190–400 nm was used as the detection wavelength. The purity of all the derivatives tested in biological essays is >95%, which meets the purity requirement according to the Journal.

#### 3.1.3. Experimental Procedures and Characterization Data for Compounds **3**, **5**, **6**, **7**, **12**, **13**, **14** and **18**

##### General Procedure for Synthesis of α-Ketiminophosphonates **3**

Following a modified literature procedure [59], to a solution of the corresponding *N*-tosyl α-aminophosphonate **9** (1 mmol) in CH2Cl2 (3 mL) was added trichloroisocyanuric acid (0.7 g, 3 mmol). The obtained suspension was stirred at 0 °C until the disappearance of the starting *N*-tosyl α-aminophosphonate **9**, as monitored by 31P NMR. The solid residue was eliminated by filtration to afford a clear solution and, then, poly(4-vinylpyridine) (0.3 g), previously dried at 100 °C overnight, was added. The suspension was stirred under reflux overnight and the reaction was then filtered and concentrated under vacuum. The resulting oily crude was purified by crystallization from diethyl ether to afford pure α-ketiminophosphonates **3**.

##### Procedure for the Synthesis of Ethyl 3-((4-methylphenyl)sulfonamido)-3-phenylpropanoate **5**

A solution of (*E*)-*N*-benzylidene-4-methylbenzenesulfonamide **1** (259.3 mg, 1 mmol) in dry CH_3_CN (3 mL) was stirred at room temperature under dry air atmosphere. To this mixture iodoacetate (0.243 mL, 2 mmol) and Et_2_Zn (8 mL, 1.0 M in hexane, 8 mmol) were added and the mixture was stirred at room temperature for 2 h. The reaction was quenched by a slow addition of a saturated aqueous solution of NH_4_Cl (3 mL), extracted with AcOEt (2 × 5 mL) and dried with anhydrous MgSO_4_. The organic layer was concentrated under reduced pressure to yield the crude product, which was purified by column chromatography in silica gel (Hexanes/AcOEt (9:1)) to afford 333 mg (96%) of **5** as a white solid. The spectroscopic data match the data reported in the literature [68].

##### Procedure for the Synthesis of Diethyl 2-((4-methylphenyl)sulfonamido)-2-phenylsuccinate **6**

A solution of ethyl (*Z*)-2-phenyl-2-(tosylimino) acetate **2** (331 mg, 1 mmol) in dry CH_3_CN (3 mL) was stirred at room temperature under dry air atmosphere. To this mixture, ethyl iodoacetate (243 μL, 2 mmol) and Me_2_Zn (3.3 mL, 1.2 M in toluene, 8 mmol) were added and the mixture was stirred at room temperature until the starting material was completely consumed. The reaction was quenched by a slow addition of a saturated aqueous solution of NH_4_Cl (3 mL), extracted with AcOEt (2 × 5 mL) and dried with anhydrous MgSO_4_. The organic layer was concentrated under reduced pressure to yield the crude product, which was purified by column chromatography in silica gel (AcOEt/Hexanes) to afford 327 mg (78%) of **6** as a white solid. M.p. (Et_2_O). 126–127 °C. ^1^H NMR (400 MHz, CDCl_3_) δ 7.27 (d, ^3^*J*_HH_ = 8.3 Hz, 2H), 7.20–7.15 (m, 3H), 7.12–7.06 (m, 2H), 7.02 (d, ^3^*J*_HH_ = 8.3 Hz, 2H), 6.3940 (s, 1H), 4.28–4.03 (m, 4H), 3.96 (d, ^3^*J*_HH_ = 16.4 Hz, 1H), 3.54 (d, ^3^*J*_HH_ = 16.4 Hz, 1H), 2.35 (s, 3H), 1.28 (t, ^3^*J*_HH_ = 7.0 Hz, 3H), 1.13 (t, ^3^*J*_HH_ = 7.0 Hz, 3H). ^13^C NMR (101 MHz, CDCl_3_) δ 171.4 (C_quat_), 170.3 (C_quat_), 142.5 (C_quat_), 139.2 (C_quat_), 135.9 (C_quat_), 129.1 (CH), 128.4 (CH), 126.8 (CH), 126.7(CH), 64.3 (C_quat_), 62.9 (CH_2_), 61.2 (CH_2_), 40.1 (CH_2_), 21.5 (CH_3_), 14.2 (CH_3_), 13.8 (CH_3_). HRMS (ESI-TOF) *m/z*: calcd for C_21_H_26_NO_6_S [M + H]^+^ 420.1481, found 420.1470.

##### General Procedure for the Aza-Reformatsky Reaction of α-Ketiminophosphonates **3**

A solution of the corresponding α-ketiminophosphonate **3** (1 mmol) in dry CH_3_CN (3 mL) was stirred at room temperature under dry air atmosphere. To this mixture, the corresponding iodoacetate (2 mmol) and Me_2_Zn (6.6 mL, 1.2 M in toluene, 8 mmol) were added and the mixture was stirred for 2h at room temperature. The reaction was quenched by a slow addition of a saturated aqueous solution of NH_4_Cl (3 mL), extracted with AcOEt (2 × 5 mL) and dried with anhydrous MgSO_4_. The organic layer was concentrated under reduced pressure to yield the crude product, which was purified by column chromatography in silica gel (AcOEt/Hexanes) [51].

*Ethyl 3-(dimethoxyphosphoryl)-3-((4-methylphenyl)sulfonamido)-3-phenylpropanoate***(7a)**. The general procedure was followed, starting form imine **3a** (367 mg, 1 mmol) to afford 414 mg (91%) of **7a** as a white solid. M.p. (CH_2_Cl_2_-hexanes). 98–99 °C. ^1^H NMR (400 MHz, CDCl_3_) δ 7.47 (d, ^3^*J*_HH_ = 8.3 Hz, 2H), 7.40–7.27 (m, 2H), 7.18 (m, 1H), 7.17–6.93 (m, 4H), 6.17 (d, ^3^*J*_PH_ = 11.2 Hz, 1H), 4.14 (q, ^3^*J*_HH_ = 7.1 Hz, 2H), 3.59 (dd, ^3^*J*_PH_ = 22.7 Hz, ^2^*J*_HH_ = 16.4 Hz, 1H), 3.46 (d, ^3^*J*_PH_ = 10.7 Hz, 3H), 3.45 (dd, ^3^*J*_PH_ = 7.5 Hz, ^2^*J*_HH_ = 16.4 Hz, 1H), 3.38 (d, ^3^*J*_PH_ = 10.5 Hz, 3H), 2.36 (s, 3H), 1.24 (t, ^3^*J*_HH_ = 7.1 Hz, 3H). ^13^C{^1^H} NMR (101 MHz, CDCl_3_) δ 170.2 (d, ^3^*J*_PC_ = 8.0 Hz, C_quat_), 143.2 (C_quat_), 139.2 (d, ^4^*J*_PC_ = 1.5 Hz, C_quat_), 134.4 (d, ^2^*J*_PC_ = 7.3 Hz, C_quat_), 129.1 (CH), 128.3 (d, ^5^*J*_PC_ = 2.9 Hz, CH), 128.2 (d, ^3^*J*_PC_ = 5.1 Hz, CH), 127.9 (d, ^4^*J*_PC_ = 2.6 Hz, 2xCH), 127.6 (CH), 62.1 (d, ^1^*J*_PC_ = 153.8 Hz, C_quat_), 61.0 (CH_2_), 54.5 (d, ^2^*J*_PC_ = 7.4 Hz, CH_3_), 54.0 (d, ^2^*J*_PC_ = 7.7 Hz, CH_3_), 38.0 (CH_2_), 21.6 (CH_3_), 14.2 (CH_3_). ^31^P NMR (120 MHz, CDCl_3_): δ 22.1. FTIR (neat) ν_max_ 3259 (N-H), 1741 (C=O), 1338 (O=S=O), 1247 (P=O), 1158 (O=S=O). HRMS (ESI-TOF) *m/z*: calcd for C_20_H_27_NO_7_PS [M + H]^+^ 456.1240, found 456.1245.

*Ethyl 3-(dimethoxyphosphoryl)-3-((4-methylphenyl)sulfonamido)-3-(p-tolyl)propanoate***(7b)**. The general procedure was followed, starting form imine **3b** (380 mg, 1 mmol) to afford 403 mg (86%) of **7b** as a white solid. M.p. (CH_2_Cl_2_-hexanes). 117–118 °C. ^1^H NMR (400 MHz, CDCl_3_) δ 7.50 (d, ^3^*J*_HH_ = 8.3 Hz, 2H), 7.24 (dd, d, ^3^*J*_HH_ = 8.6 Hz, ^4^*J*_PH_ = 2.4 Hz, 2H), 7.15 (d, ^3^*J*_HH_ = 7.9 Hz, 2H), 6.94 (d, ^3^*J*_HH_ = 8.6 Hz, 2H), 6.13 (d, ^3^*J*_PH_ = 11.3 Hz, 1H), 4.16 (qd, ^3^*J*_HH_ = 7.1, 1.3 Hz, 2H), 3.63–3.53 (m, 1H), 3.50 (d, ^3^*J*_PH_ = 10.7 Hz, 3H), 3.47–3.39 (m, 1H), 3.42 (d, ^3^*J*_PH_ = 10.7 Hz, 3H), 2.40 (s, 3H), 2.28 (s, 3H), 1.27 (t, ^3^*J*_HH_ = 7.1 Hz, 3H). ^13^C{^1^H} NMR (75 MHz, CDCl_3_) δ 170.3 (d, ^3^*J*_PC_ = 8.1 Hz, C_quat_), 143.2 (C_quat_), 139.2 (C_quat_), 138.2 (d, ^5^*J*_PC_ = 3.1 Hz, C_quat_), 131.3 (d, ^2^*J*_PC_ = 7.3 Hz, C_quat_), 129.0 (CH), 128.7 (d, ^4^*J*_PC_ = 2.7 Hz, CH), 128.1 (d, ^3^*J*_PC_ = 5.0 Hz, CH), 127.7 (CH), 61.9 (d, ^1^*J*_PC_ = 154.8 Hz, C_quat_), 61.0 (CH_2_), 54.6 (d, ^2^*J*_PC_ = 7.4 Hz, CH_3_), 54.0 (d, ^2^*J*_PC_ = 7.6 Hz, CH_3_), 37.9 (CH_2_), 21.6 (CH_3_), 21.1 (CH_3_), 14.2 (CH_3_). ^31^P NMR (120 MHz, CDCl_3_): δ 22.3. FTIR (neat) ν_max_ 3311 (N-H), 1732 (C=O), 1335 (O=S=O), 1244 (P=O), 1160 (O=S=O). HRMS (ESI-TOF) *m/z*: calcd for C_21_H_29_NO_7_PS [M + H]^+^ 470.1397, found 470.1403.

*Ethyl 3-(dimethoxyphosphoryl)-3-((4-methylphenyl)sulfonamido)-3-(m-tolyl)propanoate***(7c)**. The general procedure was followed, starting form imine **3c** (381 mg, 1 mmol) to afford 427 mg (91%) of **7c** as a pale yellow solid. M.p. (CH_2_Cl_2_-hexanes). 103–104 °C. ^1^H NMR (400 MHz, CDCl_3_) δ 7.47 (d, ^3^*J*_HH_ = 8.2 Hz, 2H), 7.14 (d, ^3^*J*_HH_ = 8.2 Hz, 2H), 7.12–6.98 (m, 4H), 6.19 (d, ^3^*J*_PH_ = 10.3 Hz, 1H), 4.18 (q, ^3^*J*_HH_ = 7.1 Hz, 2H), 3.66 (dd, ^3^*J*_PH_ = 24.4 Hz, ^2^*J*_HH_ = 16.4 Hz, 1H), 3.49 (d, ^3^*J*_PH_ = 10.7 Hz, 3H), 3.45 (m, 1H), 3.42 (d, ^3^*J*_PH_ = 10.5 Hz, 3H), 2.38 (s, 3H), 2.09 (s, 3H), 1.28 (t, ^3^*J*_HH_ = 7.1 Hz, 3H). ^13^C{^1^H} NMR (101 MHz, CDCl_3_) δ 170.4 (d, ^3^*J*_PC_ = 6.9 Hz, C_quat_), 143.2 (C_quat_), 139.4 (d, ^4^*J*_PC_ = 1.0 Hz, C_quat_), 137.5 (d, ^4^*J*_PC_ = 2.9 Hz, C_quat_), 134.0 (d, ^2^*J*_PC_ = 7.5 Hz, C_quat_), 129.5 (d, ^3^*J*_PC_ = 4.8 Hz, CH), 129.2 (d, ^5^*J*_PC_ = 3.2 Hz, CH), 129.1 (CH), 128.0 (d, ^4^*J*_PC_ = 2.7 Hz, CH), 127.6 (CH), 125.0 (d, ^3^*J*_PC_ = 5.3 Hz, CH), 62.2 (d, ^1^*J*_PC_ = 153.8 Hz, C_quat_), 61.1 (CH_2_), 54.6 (d, ^2^*J*_PC_ = 7.4 Hz, CH_3_), 54.1 (d, ^2^*J*_PC_ = 7.7 Hz, CH_3_), 38.3 (CH_2_), 21.6 (CH_3_), 21.5 (CH_3_), 14.3 (CH_3_). ^31^P NMR (120 MHz, CDCl_3_): δ 22.4. FTIR (neat) ν_max_ 3276 (N-H), 1735 (C=O), 1338 (O=S=O), 1241 (P=O), 1157 (O=S=O). HRMS (ESI-TOF) *m/z*: calcd for C_21_H_29_NO_7_PS [M + H]^+^ 470.1397, found 470.1398.

*Ethyl 3-(dimethoxyphosphoryl)-3-((4-methylphenyl)sulfonamido)-3-(4-((trichloromethyl)thio)phenyl)propanoate***(7d)**. The general procedure was followed, starting form imine **3d** (516 mg, 1 mmol) to afford 525 mg (87%) of **7d** as a pale yellow solid. M.p. (CH_2_Cl_2_-hexanes). 123–124 °C. ^1^H NMR (400 MHz, CDCl_3_) δ 7.57 (d, ^3^*J*_HH_ = 8.3 Hz, 2H), 7.53–7.46 (m, 4H), 7.17 (d, ^3^*J*_HH_ = 8.1 Hz, 2H), 6.26 (d, ^3^*J*_PH_ = 10.5 Hz, 1H), 4.15 (m, 2H), 3.63 (dd, ^3^*J*_PH_ = 23.5 Hz, ^2^*J*_HH_ = 16.4 Hz, 1H), 3.49 (d, ^3^*J*_PH_ = 10.8 Hz, 3H), 3.46 (m, 1H), 3.49 (d, ^3^*J*_PH_ = 10.6 Hz, 3H), 2.38 (s, 3H), 1.26 (t, ^3^*J*_HH_ = 7.1 Hz, 3H). ^13^C{^1^H} NMR (101 MHz, CDCl_3_) δ 169.9 (d, ^3^*J*_PC_ = 7.1 Hz, C_quat_), 143.7 (C_quat_), 139.1 (d, ^4^*J*_PC_ = 1.2 Hz, C_quat_), 138.8 (d, ^2^*J*_PC_ = 7.3 Hz, C_quat_), 136.5 (d, ^4^*J*_PC_ = 2.8 Hz, CH), 130.8 (d, ^5^*J*_PC_ = 3.6 Hz, C_quat_), 129.4 (CH), 129.2 (d, ^3^*J*_PC_ = 4.9 Hz, CH), 127.5 (CH), 98.5 (d, ^7^*J*_PC_ = 3.2 Hz, C_quat_), 62.3 (d, ^1^*J*_PC_ = 152.2 Hz, C_quat_), 61.2 (CH_2_), 54.8 (d, ^2^*J*_PC_ = 7.4 Hz, CH_3_), 54.4 (d, ^2^*J*_PC_ = 7.6 Hz, CH_3_), 38.3 (CH_2_), 21.7 (CH_3_), 14.2 (CH_3_). ^31^P NMR (120 MHz, CDCl_3_): δ 21.5. FTIR (neat) ν_max_ 3291 (N-H), 1733 (C=O), 1331 (O=S=O), 1258 (P=O), 1159 (O=S=O). HRMS (ESI-TOF) *m/z*: calcd for C_21_H_26_Cl_3_NO_7_PS_2_ [M + H]^+^ 605.9919, found 605.9928. 

*Ethyl 3-(4-bromophenyl)-3-(dimethoxyphosphoryl)-3-((4-methylphenyl)sulfonamido)propanoate***(7e).** The general procedure was followed, starting form imine **3e** (445 mg, 1 mmol) to afford 485 mg (91%) of **7e** as a white solid. M.p. (CH_2_Cl_2_-hexanes). 138–139 °C. ^1^H NMR (400 MHz, CDCl_3_) δ 7.38 (d, ^3^*J*_HH_ = 8.3 Hz, 2H), 7.18–7.12 (m, 4H), 7.07 (d, ^3^*J*_HH_ = 8.3 Hz, 2H), 6.24 (d, ^3^*J*_PH_ = 10.1 Hz, 1H), 4.07 (m, 2H), 3.49 (m, 1H), 3.47 (d, ^3^*J*_PH_ = 10.8 Hz, 3H), 3.39 (d, ^3^*J*_PH_ = 10.6 Hz, 3H), 3.32 (dd, ^2^*J*_HH_ = 16.4 Hz, ^3^*J*_PH_ = 8.0 Hz, 1H), 2.32 (s, 3H), 1.18 (t, ^3^*J*_HH_ = 7.1 Hz, 3H). ^13^C{^1^H} NMR (101 MHz, CDCl_3_) δ 169.6 (d, ^3^*J*_PC_ = 7.7 Hz, C_quat_), 143.3 (C_quat_), 138.8 (m, C_quat_), 133.4 (d, ^2^*J*_PC_ = 6.7 Hz, C_quat_), 130.6 (d, ^4^*J*_PC_ = 2.6 Hz, CH), 129.8 (d, ^3^*J*_PC_ = 5.1 Hz, CH), 129.0 (CH), 127.3 (CH), 122.5 (d, ^5^*J*_PC_ = 3.6 Hz, C_quat_), 61.6 (d, ^1^*J*_PC_ = 154.3 Hz, C_quat_), 60.8 (CH_2_), 54.4 (d, ^2^*J*_PC_ = 7.4 Hz, CH_3_), 54.0 (d, ^2^*J*_PC_ = 7.7 Hz, CH_3_), 37.6 (CH_2_), 21.36 (CH_3_), 14.0 (CH_3_). ^31^P NMR (120 MHz, CDCl_3_): δ 21.7. FTIR (neat) ν_max_ 3293 (N-H), 1729 (C=O), 1337 (O=S=O), 1245 (P=O), 1154 (O=S=O). HRMS (ESI-TOF) *m/z*: calcd for C_20_H_26_BrNO_7_PS [M + H]^+^ 534.0345, found 534.0311.

*Ethyl 3-(4-chlorophenyl)-3-(dimethoxyphosphoryl)-3-((4-methylphenyl)sulfonamido)propanoate***(7f).** The general procedure was followed, starting form imine **3f** (401 mg, 1 mmol) to afford 430 mg (88%) of **7f** as a pale yellow solid. M.p. (CH_2_Cl_2_-hexanes). 133–134 °C. ^1^H NMR (400 MHz, CDCl_3_) δ 7.47 (d, ^3^*J*_HH_ = 8.1 Hz, 2H), 7.29 (dd, ^3^*J*_HH_ = 8.7 Hz, ^4^*J*_PH_ = 2.4 Hz, 2H), 7.16 (d, ^3^*J*_HH_ = 8.0 Hz, 2H), 7.08 (d, ^3^*J*_HH_ = 8.6 Hz, 2H), 6.18 (d, ^3^*J*_PH_ = 10.3 Hz, 1H), 4.17 (q, ^3^*J*_HH_ = 7.2 Hz, 2H), 3.62 (m, 1H), 3.55 (d, ^3^*J*_PH_ = 10.5 Hz, 3H), 3.47 (d, ^3^*J*_PH_ = 10.5 Hz, 3H), 3.40 (dd, ^2^*J*_HH_ = 16.1 Hz, ^3^*J*_PH_ = 7.5 Hz, 1H), 2.41 (s, 3H), 1.28 (t, ^3^*J*_HH_ = 7.2 Hz, 3H). ^13^C{^1^H} NMR (75 MHz, CDCl_3_) δ 170.0 (d, ^3^*J*_PC_ = 7.2 Hz, C_quat_), 143.5 (C_quat_), 139.0 (C_quat_), 134.5 (d, ^5^*J*_PC_ = 3.5 Hz, C_quat_), 133.0 (d, ^2^*J*_PC_ = 7.0 Hz, C_quat_), 129.7 (d, ^3^*J*_PC_ = 5.1 Hz, CH), 129.2 (CH), 128.0 (d, ^4^*J*_PC_ = 2.7 Hz, CH), 127.6 (CH), 61.7 (d, ^1^*J*_PC_ = 154.2 Hz, C_quat_), 61.1 (CH_2_), 54.7 (d, ^2^*J*_PC_ = 7.3 Hz, CH_3_), 54.2 (d, ^2^*J*_PC_ = 7.5 Hz, CH_3_), 38.0 (CH_2_), 21.6 (CH_3_), 14.2 (CH_3_). ^31^P NMR (120 MHz, CDCl_3_): δ 21.9. FTIR (neat) ν_max_ 3256 (N-H), 1732 (C=O), 1335 (O=S=O), 1246 (P=O), 1160 (O=S=O). HRMS (ESI-TOF) *m/z*: calcd for C_20_H_26_ClNO_7_PS [M + H]^+^ 490.0851, found 490.0856. 

*Ethyl 3-(3-chlorophenyl)-3-(dimethoxyphosphoryl)-3-((4-methylphenyl)sulfonamido)propanoate***(7g)**. The general procedure was followed, starting form imine **3g** (401 mg, 1 mmol) to afford 440 mg (90%) of **7g** as a pale yellow solid. M.p. (CH_2_Cl_2_-hexanes). 88–89 °C. ^1^H NMR (400 MHz, CDCl_3_) δ 7.46 (d, ^3^*J*_HH_ = 8.3 Hz, 2H), 7.27–7.22 (m, 2H), 7.19–7.09 (m, 4H), 6.20 (d, ^3^*J*_PH_ = 9.4 Hz, 1H), 4.20 (q, ^3^*J*_HH_ = 7.1 Hz, 2H), 3.65 (dd, ^3^*J*_PH_ = 24.9 Hz, ^3^*J*_HH_ = 16.2 Hz, 1H), 3.54 (d, ^3^*J*_PH_ = 10.7 Hz, 3H), 3.51 (d, ^3^*J*_PH_ = 10.6 Hz, 3H), 3.42 (dd, ^3^*J*_HH_ = 16.2 Hz, ^3^*J*_PH_ = 7.6 Hz, 1H), 2.40 (s, 3H), 1.29 (t, ^3^*J*_HH_ = 7.1 Hz, 3H). ^13^C{^1^H} NMR (101 MHz, CDCl_3_) δ 170.1 (d, ^3^*J*_PC_ = 6.1 Hz, C_quat_), 143.7 (C_quat_), 139.0 (C_quat_), 136.5 (d, ^2^*J*_PC_ = 7.7 Hz, C_quat_), 134.1 (d, ^4^*J*_PC_ = 3.3 Hz, C_quat_), 129.4 (CH), 129.2 (d, ^4^*J*_PC_ = 2.9 Hz, CH), 129.2 (d, ^3^*J*_PC_ = 4.9 Hz, CH), 128.6 (d, ^5^*J*_PC_ = 2.9 Hz, CH), 127.5 (CH), 126.2 (d, ^3^*J*_PC_ = 5.1 Hz, CH), 62.0 (d, ^1^*J*_PC_ = 153.2 Hz, C_quat_), 61.2 (CH_2_), 54.7 (d, ^2^*J*_PC_ = 7.3 Hz, CH_3_), 54.4 (d, ^2^*J*_PC_ = 7.6 Hz, CH_3_), 38.3 (CH_2_), 21.7 (CH_3_), 14.3 (CH_3_). ^31^P NMR (120 MHz, CDCl_3_): δ 21.9. FTIR (neat) ν_max_ 3276 (N-H), 1732 (C=O), 1341 (O=S=O), 1238 (P=O), 1161 (O=S=O). HRMS (ESI-TOF) *m/z*: calcd for C_20_H_26_ClNO_7_PS [M + H]^+^ 490.0851, found 490.0857.

*Ethyl 3-(3,4-dichlorophenyl)-3-(dimethoxyphosphoryl)-3-((4-methylphenyl)sulfonamido)propanoate***(7h)**. The general procedure was followed, starting form imine **3h** (435 mg, 1 mmol) to afford 450 mg (86%) of **7h** as a white solid. M.p. (CH_2_Cl_2_-hexanes). 109–110 °C. ^1^H NMR (400 MHz, CDCl_3_) δ 7.43 (d, ^3^*J*_HH_ = 8.2 Hz, 2H), 7.33 (s, 1H), 7.23–7.18 (m, 2H), 7.15 (d, ^3^*J*_HH_ = 8.2 Hz, 2H), 6.24 (d, ^3^*J*_PH_ = 9.3 Hz, 1H), 4.17 (q, ^3^*J*_HH_ = 7.5 Hz, 2H), 3.59 (m, 1H), 3.58 (d, ^3^*J*_HH_ = 10.8 Hz, 3H), 3.54 (d, ^3^*J*_HH_ = 10.7 Hz, 3H), 3.36 (dd, ^2^*J*_HH_ = 16.2 Hz, ^3^*J*_PH_ = 8.3 Hz, 1H), 2.38 (s, 3H), 1.27 (t, ^3^*J*_HH_ = 7.5Hz, 3H). ^13^C{^1^H} NMR (101 MHz, CDCl_3_) δ 169.7 (d, ^3^*J*_PC_ = 6.4 Hz, C_quat_), 143.9 (C_quat_), 138.7 (d, ^4^*J*_PC_ = 1.4 Hz, C_quat_), 134.7 (d, ^2^*J*_PC_ = 6.6 Hz, C_quat_), 132.7 (d, ^5^*J*_PC_ = 3.7 Hz, C_quat_), 132.1 (d, ^4^*J*_PC_ = 2.9 Hz, C_quat_), 130.9 (d, ^3^*J*_PC_ = 5.1 Hz, CH), 129.7 (d, ^4^*J*_PC_ = 2.6 Hz, CH), 129.4 (CH), 127.4 (d, ^3^*J*_PC_ = 5.1 Hz, CH), 127.3 (CH), 61.4 (d, ^1^*J*_PC_ = 153.7 Hz, C_quat_), 61.2 (CH_2_), 54.7 (d, ^2^*J*_PC_ = 7.3 Hz, CH_3_), 54.5 (d, ^2^*J*_PC_ = 7.6 Hz, CH_3_), 38.0 (CH_2_), 21.6 (CH_3_), 14.2 (CH_3_). ^31^P NMR (120 MHz, CDCl_3_): δ 21.7. FTIR (neat) ν_max_ 3251 (N-H), 1738 (C=O), 1338 (O=S=O), 1261 (P=O), 1160 (O=S=O). HRMS (ESI-TOF) *m/z*: calcd for C_20_H_25_Cl_2_NO_7_PS [M + H]^+^ 524.0461, found 524.0465.

*Ethyl 3-(3-chloro-4-methoxyphenyl)-3-(dimethoxyphosphoryl)-3-((4-methylphenyl)sulfonamido)propanoate***(7i)**. The general procedure was followed, starting form imine **3i** (431 mg, 1 mmol) to afford 420 mg (81%) of **7i** as a white solid. M.p. (CH_2_Cl_2_-hexanes). 153–154 °C. ^1^H NMR (400 MHz, CDCl_3_) δ 7.44 (d, ^3^*J*_HH_ = 8.3 Hz, 2H), 7.24–7.18 (m, 2H), 7.15 (d, ^3^*J*_HH_ = 8.3 Hz, 2H), 6.71 (d, ^3^*J*_HH_ = 8.7 Hz, 1H), 6.17 (d, ^3^*J*_PH_ = 9.0 Hz, 1H), 4.20 (q, ^3^*J*_HH_ = 7.1 Hz, 2H), 3.86 (s, 3H), 3.62 (dd, ^3^*J*_PH_ = 25.6 Hz, ^2^*J*_HH_ = 16.1 Hz, 1H), 3.55 (d, ^3^*J*_PH_ = 10.7 Hz, 3H), 3.52 (d, ^3^*J*_PH_ = 10.7 Hz, 3H), 3.36 (dd, ^2^*J*_HH_ = 16.1 Hz, ^3^*J*_PH_ = 7.6 Hz, 1H), 2.39 (s, 3H), 1.30 (t, ^3^*J*_HH_ = 7.1 Hz, 3H). ^13^C{^1^H} NMR (101 MHz, CDCl_3_) δ 170.1 (d, ^3^*J*_PC_ = 5.9 Hz, C_quat_), 154.9 (d, ^5^*J*_PC_ = 2.7 Hz, C_quat_), 143.7 (C_quat_), 138.9 (d, ^4^*J*_PC_ = 1.7 Hz, C_quat_),130.9 (d, ^3^*J*_PC_ = 4.8 Hz, CH), 129.3 (CH), 127.7 (d, ^3^*J*_PC_ = 5.5 Hz, CH), 127.5 (CH), 126.8 (d, ^2^*J*_PC_ = 7.4 Hz, C_quat_), 122.0 (d, ^4^*J*_PC_ = 3.0 Hz, C_quat_), 111.0 (d, ^4^*J*_PC_ = 2.6 Hz, CH), 61.3 (d, ^1^*J*_PC_ = 155.4 Hz, C_quat_), 61.2 (CH_2_), 56.2 (CH_3_), 54.7 (d, ^2^*J*_PC_ = 7.3 Hz, CH_3_), 54.3 (d, ^2^*J*_PC_ = 7.7 Hz, CH_3_), 38.2 (CH_2_), 21.7 (CH_3_), 14.3 (CH_3_). ^31^P NMR (120 MHz, CDCl_3_): δ 22.2. FTIR (neat) ν_max_ 3275 (N-H), 1732 (C=O), 1333 (O=S=O), 1263 (P=O), 1158 (O=S=O). HRMS (ESI-TOF) *m/z*: calcd for C_21_H_28_ ClNO_8_PS [M + H]^+^ 520.0956, found 520.0959.

*Ethyl 3-(dimethoxyphosphoryl)-3-(4-fluorophenyl)-3-((4-methylphenyl)sulfonamido)propanoate***(7j)**. The general procedure was followed, starting form imine **3j** (417 mg, 1 mmol) to afford 421 mg (89%) of **7j** as a white solid. M.p. (CH_2_Cl_2_-hexanes). 111–112 °C. ^1^H NMR (300 MHz, CDCl_3_) δ 7.47 (d, ^3^*J*_HH_ = 8.5 Hz, 2H), 7.34 (m, 2H), 7.16 (d, ^3^*J*_HH_ = 8.1 Hz, 2H), 6.81 (t, ^3^*J*_HH_ = 8.7 Hz, 2H), 6.19 (d, ^3^*J*_PH_ = 10.0 Hz, 1H), 4.18 (q, ^3^*J*_HH_ = 7.1 Hz, 2H), 3.62 (m, 1H), 3.54 (d, ^3^*J*_PH_ = 10.5 Hz, 3H), 3.46 (d, ^3^*J*_PH_ = 10.5 Hz, 3H), 3.40 (m, 1H), 2.40 (s, 3H), 1.28 (t, ^3^*J*_HH_ = 7.2 Hz, 3H). ^13^C{^1^H} NMR (75 MHz, CDCl_3_) δ 170.1 (d, ^3^*J*_PC_ = 7.0 Hz, C_quat_), 162.5 (dd, ^1^*J*_FC_ = 248.9 Hz, ^5^*J*_PC_ = 3.2 Hz, C_quat_), 143.5 (C_quat_), 139.1 (C_quat_), 130.3 (dd, ^3^*J*_FC_ = 8.3 Hz, ^3^*J*_PC_ = 5.0 Hz, CH), 130.1 (dd, ^2^*J*_PC_ = 6.9 Hz, ^4^*J*_FC_ = 3.3 Hz, C_quat_), 129.2 (CH), 127.6 (CH), 114.8 (dd, ^2^*J*_FC_ = 21.6 Hz, ^4^*J*_PC_ = 2.7 Hz, CH), 61.7 (d, ^1^*J*_PC_ = 155.0 Hz, C_quat_), 61.2 (CH_2_), 54.7 (d, ^2^*J*_PC_ = 7.4 Hz, CH_3_), 54.2 (d, ^2^*J*_PC_ = 7.7 Hz, CH_3_), 38.2 (CH_2_), 21.6 (CH_3_), 14.2 (CH_3_). ^31^P NMR (120 MHz, CDCl_3_): δ 22.2. ^19^F NMR (282 MHz, CDCl_3_) δ -113.8. FTIR (neat) ν_max_ 3264 (N-H), 1735 (C=O), 1335 (O=S=O), 1241 (P=O), 1161 (O=S=O). HRMS (ESI-TOF) *m/z*: calcd for C_20_H_26_FNO_7_PS [M + H]^+^ 474.1146, found 474.1148.

*Ethyl 3-(dimethoxyphosphoryl)-3-(3-fluorophenyl)-3-((4-methylphenyl)sulfonamido)propanoate***(7k)**. The general procedure was followed, starting form imine **3k** (417 mg, 1 mmol) to afford 421 mg (89%) of **7k** as a white solid. M.p. (CH_2_Cl_2_-hexanes). 111–112 °C. ^1^H NMR (400 MHz, CDCl_3_) δ 7.48 (d, ^3^*J*_HH_ = 8.3 Hz, 2H), 7.17–7.09 (m, 4H), 7.01 (m, 1H), 6.90 (m, 1H), 6.22 (br s, 1H), 4.15 (q, ^3^*J*_HH_ = 7.1 Hz, 2H), 3.59 (dd, ^3^*J*_PH_ = 23.5 Hz, ^2^*J*_HH_ = 16.4 Hz, 1H), 3.51 (d, ^3^*J*_PH_ = 10.7 Hz, 3H), 3.47 (d, ^3^*J*_PH_ = 10.7 Hz, 3H), 3.40 (dd, ^2^*J*_HH_ = 16.4 Hz, ^3^*J*_PH_ = 7.5 Hz, 1H), 2.37 (s, 3H), 1.25 (t, ^3^*J*_HH_ = 7.1 Hz, 3H). ^13^C{^1^H} NMR (101 MHz, CDCl_3_) δ 169.9 (d, ^3^*J*_PC_ = 7.3 Hz, C_quat_), 162.2 (dd, ^1^*J*_FC_ = 246.1 Hz, ^4^*J*_PC_ = 2.9 Hz, C_quat_), 143.6 (C_quat_), 139.0 (d, ^4^*J*_PC_ = 1.4 Hz, C_quat_), 137.1 (t, ^2^*J*_PC_ = ^3^*J*_FC_ = 7.1 Hz, C_quat_), 129.4 (dd, ^3^*J*_FC_ = 8.1 Hz, ^4^*J*_PC_ = 2.8 Hz, CH), 129.2 (CH), 127.5 (CH), 123.7 (dd, ^3^*J*_PC_ = 5.2 Hz, ^4^*J*_FC_ = 2.9 Hz, CH), 116.0 (dd, ^2^*J*_FC_ = 24.1 Hz, ^3^*J*_PC_ = 4.9 Hz, CH), 115.3 (dd, ^2^*J*_FC_ = 21.0 Hz, ^5^*J*_PC_ = 3.0 Hz, CH), 61.9 (dd, ^1^*J*_PC_ = 153.6 Hz, ^4^*J*_FC_ = 1.9 Hz, C_quat_), 61.1 (CH_2_), 54.6 (d, ^2^*J*_PC_ = 7.4 Hz, CH_3_), 54.3 (d, ^2^*J*_PC_ = 7.6 Hz, CH_3_), 38.1 (CH_2_), 21.6 (CH_3_), 14.2 (CH_3_). ^31^P NMR (120 MHz, CDCl_3_): δ 21.7. ^19^F NMR (282 MHz, CDCl_3_) δ −113.0. FTIR (neat) ν_max_ 3281 (N-H), 1732 (C=O), 1333 (O=S=O), 1244 (P=O), 1157 (O=S=O). HRMS (ESI-TOF) *m/z*: calcd for C_20_H_26_FNO_7_PS [M + H]^+^ 474.1146, found 474.1155.

*Ethyl 3-(dimethoxyphosphoryl)-3-(2-fluorophenyl)-3-((4-methylphenyl)sulfonamido)propanoate***(7l)**. The general procedure was followed, starting form imine **3l** (417 mg, 1 mmol) to afford 440 mg (93%) of **7l** as a pale yellow solid. M.p. (CH_2_Cl_2_-hexanes). 140–141 °C. ^1^H NMR (400 MHz, CDCl_3_) δ 7.58 (d, ^3^*J*_HH_ = 8.3 Hz, 2H), 7.48–7.40 (m, 1H), 7.23–7.19 (m, 1H), 7.16 (d, ^3^*J*_HH_ = 8.3 Hz, 2H), 6.99 (m, 1H), 6.80 (dd, ^3^*J*_FH_ = 12.8 Hz, ^3^*J*_HH_ = 8.1 Hz, 1H), 6.21 (d, ^3^*J*_PH_ = 13.1 Hz, 1H), 4.17–4.01 (m, 2H), 3.63 (d, ^3^*J*_PH_ = 10.8 Hz, 3H), 3.60 (m, 2H), 3.54 (d, ^3^*J*_PH_ = 10.7 Hz, 3H),2.38 (s, 3H), 1.23 (t, ^3^*J*_HH_ = 7.1 Hz, 3H). ^13^C{^1^H} NMR (101 MHz, CDCl_3_) δ 170.2 (d, ^3^*J*_PC_ = 10.8 Hz, C_quat_), 160.6 (dd, ^1^*J*_FC_ = 249.9 Hz, ^3^*J*_PC_ = 5.8 Hz, C_quat_), 143.3 (C_quat_), 138.8 (d, ^4^*J*_PC_ = 1.3 Hz, C_quat_), 130.4 (dd, ^3^*J*_FC_ = 9.4, ^3^*J*_PC_ = 2.8 Hz, CH), 130.3 (dd, ^3^*J*_FC_ = 4.8 Hz, ^5^*J*_PC_ = 2.9 Hz, CH), 129.2 (CH), 127.6 (CH), 123.9 (dd, ^4^*J*_FC_ = 3.3 Hz, ^4^*J*_PC_ = 2.5 Hz, CH), 123.2 (m, C_quat_), 116.4 (dd, ^2^*J*_FC_ = 24.6 Hz, ^4^*J*_PC_ = 2.5 Hz, CH), 61.4 (dd, ^1^*J*_PC_ = 154.0, ^3^*J*_FC_ = 3.4 Hz, C_quat_), 60.9 (CH_2_), 54.7 (d, ^2^*J*_PC_ = 7.6 Hz, CH_3_), 54.5 (d, ^2^*J*_PC_ = 7.4 Hz, CH_3_), 38.6 (d, ^4^*J*_FC_ = 6.8 Hz, CH_2_), 21.6 (CH_3_), 14.2 (CH_3_). ^31^P NMR (120 MHz, CDCl_3_): δ 22.1. ^19^F NMR (282 MHz, CDCl_3_) δ −107.6. FTIR (neat) ν_max_ 3236 (N-H), 1755 (C=O), 1328 (O=S=O), 1239 (P=O), 1160 (O=S=O). HRMS (ESI-TOF) *m/z*: calcd for C_20_H_26_FNO_7_PS [M + H]^+^ 474.1146, found 474.1126. The spectroscopic data match the data reported in the literature [51].

*Ethyl 3-(2,4-difluorophenyl)-3-(dimethoxyphosphoryl)-3-((4-methylphenyl)sulfonamido)propanoate***(7m)**. The general procedure was followed, starting form imine **3m** (403 mg, 1 mmol) to afford 457 mg (93%) of **7m** as a white solid. M.p. (CH_2_Cl_2_-hexanes). 135–136 °C. ^1^H NMR (400 MHz, CDCl_3_) δ 7.55 (d, ^3^*J*_HH_ = 8.3 Hz, 2H), 7.39 (m, 1H), 7.15 (d, ^3^*J*_HH_ = 8.3 Hz, 2H), 6.68 (m, 1H), 6.52 (m, 1H), 6.25 (br d, ^3^*J*_PH_ = 8.9 Hz, 1H), 4.09 (m, 2H), 3.65 (d, ^3^*J*_PH_ = 10.8 Hz, 3H), 3.57 (d, ^3^*J*_PH_ = 10.7 Hz, 3H), 3.57–3.47 (m, 2H), 2.37 (s, 3H), 1.22 (t, ^3^*J*_HH_ = 7.1 Hz, 3H). ^13^C{^1^H} NMR (101 MHz, CDCl_3_) δ 170.0 (d, ^3^*J*_PC_ = 10.9 Hz, C_quat_), 162.9 (ddd, ^1^*J*_FC_ = 251.3 Hz, ^3^*J*_FC_ = 12.7 Hz, ^5^*J*_PC_ = 2.9 Hz, C_quat_), 160.7 (ddd, ^1^*J*_FC_ = 252.9 Hz, ^3^*J*_FC_ = 11.7 Hz, ^3^*J*_PC_ = 5.7 Hz, C_quat_), 143.5 (C_quat_), 138.6 (C_quat_), 131.4 (m, CH), 129.2 (CH), 127.5 (CH), 119.4 (m, C_quat_), 110.8 (br d, ^2^*J*_FC_ = 21 Hz, CH), 104.5 (ddd, ^2^*J*_FC_ = 27.9 Hz, ^2^*J*_FC_ = 25.3 Hz, ^4^*J*_PC_ = 2.5 Hz, CH), 60.9 (CH_2_), 60.9 (dd, ^1^*J*_PC_ = 154.6 Hz, ^3^*J*_FC_ = 3.7 Hz, C_quat_), 54.6 (br d, ^2^*J*_PC_ = 7.6 Hz, CH_3_), 54.6 (br d, ^2^*J*_PC_ = 7.3 Hz, CH_3_), 38.5 (d, ^4^*J*_FC_ = 6.9 Hz, CH_2_), 21.6 (CH_3_), 14.1 (CH_3_). ^31^P NMR (120 MHz, CDCl_3_): δ 22.0. ^19^F NMR (282 MHz, CDCl_3_) δ −103.0, −110.3 ppm. FTIR (neat) ν_max_ 3248 (N-H), 1741 (C=O), 1335 (O=S=O), 1247 (P=O), 1155 (O=S=O). HRMS (ESI-TOF) *m/z*: calcd for C_20_H_25_F_2_NO_7_PS [M + H]^+^ 492.1052, found 492.1058.

*Ethyl 3-(3,4-difluorophenyl)-3-(dimethoxyphosphoryl)-3-((4-methylphenyl)sulfonamido)propanoate***(7n)**. The general procedure was followed, starting form imine **3n** (403 mg, 1 mmol) to afford 447 mg (91%) of **7n** as a white solid. M.p. (CH_2_Cl_2_-hexanes). 98–99 °C. ^1^H NMR (400 MHz, CDCl_3_) δ 7.48 (d, ^3^*J*_HH_ = 8.0 Hz, 2H), 7.19 (d, ^3^*J*_HH_ = 8.0 Hz, 2H), 7.17–7.07 (m, 2H), 6.95 (q, ^3^*J*_HH_ = 9.2 Hz, 1H), 6.21 (d, ^3^*J*_PH_ = 9.6 Hz, 1H), 4.19 (q, ^3^*J*_HH_ = 7.1 Hz, 2H), 3.61 (m, 1H), 3.59 (d, ^3^*J*_PH_ = 11.1 Hz, 3H), 3.53 (d, ^3^*J*_PH_ = 10.6 Hz, 3H), 3.36 (dd, ^2^*J*_HH_ = 16.1 Hz, ^3^*J*_PH_ = 7.9 Hz, 1H), 2.41 (s, 3H), 1.29 (t, ^3^*J*_HH_ = 7.2 Hz, 3H). ^13^C{^1^H} NMR (101 MHz, CDCl_3_) δ 169.8 (d, ^3^*J*_PC_ = 6.8 Hz, C_quat_), 150.2 (ddd, ^1^*J*_FC_ = 250.6 Hz, ^2^*J*_FC_ = 11.8 Hz, ^5^*J*_PC_ = 3.1 Hz, C_quat_), 149.5 (ddd, ^1^*J*_FC_ = 246.0 Hz, ^2^*J*_FC_ = 10.6 Hz, ^4^*J*_PC_ = 3.0 Hz, C_quat_), 143.9 (C_quat_), 139.0 (d, ^4^*J*_PC_ = 1.3 Hz, C_quat_), 131.6 (m, C_quat_), 129.3 (CH), 127.5 (CH), 124.4 (m, CH), 118.4 (dd, ^2^*J*_FC_ = 19.6 Hz, ^3^*J*_PC_ = 4.8 Hz, CH), 116.5 (dd, ^2^*J*_FC_ = 17.4 Hz, ^4^*J*_PC_ = 2.7 Hz, CH), 61.5 (d, ^1^*J*_PC_ = 154.6 Hz, C_quat_), 61.3 (CH_2_), 54.7 (d, ^2^*J*_PC_ = 7.3 Hz, CH_3_), 54.4 (d, ^2^*J*_PC_ = 7.6 Hz, CH_3_), 38.2 (CH_2_), 21.6 (CH_3_), 14.2 (CH_3_). ^31^P NMR (120 MHz, CDCl_3_): δ 21.7. ^19^F NMR (282 MHz, CDCl_3_) δ −137.6, −138.0. FTIR (neat) ν_max_ 3262 (N-H), 1738 (C=O), 1338 (O=S=O), 1249 (P=O), 1163 (O=S=O). HRMS (ESI-TOF) *m/z*: calcd for C_20_H_25_F_2_NO_7_PS [M + H]^+^ 492.1052, found 492.1060.

*Ethyl 3-(dimethoxyphosphoryl)-3-((4-methylphenyl)sulfonamido)-3-(perfluorophenyl)propanoate***(7o)**. The general procedure was followed, starting form imine **3o** (457 mg, 1 mmol) to afford 501 mg (92%) of **7o** as a colorless oil. ^1^H NMR (400 MHz, CDCl_3_) δ 7.52 (d, ^3^*J*_HH_ = 8.3 Hz, 2H), 7.16 (d, ^3^*J*_HH_ = 8.3 Hz, 2H), 6.23 (d, ^3^*J*_PH_ = 10.5 Hz, 1H), 4.18–4.07 (m, 2H), 3.98 (m, 1H), 3.80 (d, ^3^*J*_PH_ = 10.9 Hz, 3H), 3.75 (d, ^3^*J*_PH_ = 10.9 Hz, 3H), 3.45 (m, 1H), 2.36 (s, 3H), 1.24 (t, ^3^*J*_HH_ = 7.1 Hz, 3H). ^13^C{^1^H} NMR (101 MHz, CDCl_3_) δ 170.1 (d, ^3^*J*_PC_ = 8.4 Hz, C_quat_), 145.9 (m, C_quat_), 144.1 (C_quat_), 140.8 (m, C_quat_), 137.7 (d, ^4^*J*_PC_ = 1.9 Hz, C_quat_), 137.5 (m, C_quat_), 129.2 (CH), 127.1 (CH), 111.2 (m, C_quat_), 61.0 (CH_2_), 60.0 (d, ^1^*J*_PC_ = 153.3 Hz, C_quat_), 55.7 (d, ^2^*J*_PC_ = 7.2 Hz, CH_3_), 54.9 (d, ^2^*J*_PC_ = 7.8 Hz, CH_3_), 40.3 (d, ^4^*J*_FC_ = 5.3 Hz, CH_2_), 21.4 (CH_3_), 14.1 (CH_3_). ^31^P NMR (120 MHz, CDCl_3_): δ 21.7. ^19^F NMR (282 MHz, CDCl_3_) δ −135.0, −154.3, −162.8. FTIR (neat) ν_max_ 3284 (N-H), 1741 (C=O), 1333 (O=S=O), 1254 (P=O), 1166 (O=S=O). HRMS (ESI-TOF) *m/z*: calcd for C_20_H_22_F_5_NO_7_PS [M + H]^+^ 546.0769, found 546.0781. 

*Ethyl 3-(dimethoxyphosphoryl)-3-((4-methylphenyl)sulfonamido)-3-(4-(trifluoromethyl)phenyl)propanoate***(7p)**. The general procedure was followed, starting form imine **3p** (435 mg, 1 mmol) to afford 455 mg (87%) of **7p** as a white solid. M.p. (CH_2_Cl_2_-hexanes). 114–115 °C. ^1^H NMR (400 MHz, CDCl_3_) δ 7.47 (^3^*J*_HH_ = 8.8 Hz, ^3^*J*_FH_ = 2.3 Hz, 2H), 7.42 (d, ^3^*J*_HH_ = 8.4 Hz, 2H), 7.34 (d, ^3^*J*_HH_ = 8.4 Hz, 2H), 7.12 (d, ^3^*J*_HH_ = 8.2 Hz, 2H), 6.24 (d, ^3^*J*_PH_ = 10.1 Hz, 1H), 4.18 (qd, ^3^*J*_HH_ = 7.2 Hz, ^3^*J*_FH_ = 1.7 Hz, 2H), 3.68 (dd, ^3^*J*_PH_ = 23.8 Hz, ^2^*J*_HH_ = 16.2 Hz, 1H), 3.58 (d, ^3^*J*_PH_ = 10.8 Hz, 3H), 3.49 (d, ^3^*J*_PH_ = 10.8 Hz, 3H), 3.44 (dd, ^3^*J*_PH_ = 8.2 Hz, ^2^*J*_HH_ = 16.2 Hz, 1H), 2.39 (s, 3H), 1.28 (t, ^3^*J*_HH_ = 7.1 Hz, 3H). ^13^C{^1^H} NMR (75 MHz, CDCl_3_) δ 170.0 (d, ^3^*J*_PC_ = 6.8 Hz, C_quat_), 143.7 (C_quat_), 138.9 (C_quat_), 138.6 (d, ^2^*J*_PC_ = 6.6 Hz, C_quat_), 130.4 (dq, ^2^*J*_FC_ = 32.8 Hz, ^5^*J*_PC_ = 3.0 Hz, C_quat_), 129.3 (CH), 128.8 (d, ^3^*J*_PC_ = 5.0 Hz, CH), 127.5 (CH), 124.7 (m, CH), 123.9 (q, ^1^*J*_FC_ = 272.4 Hz, C_quat_), 62.0 (d, ^1^*J*_PC_ = 152.8 Hz, C_quat_), 61.3 (CH_2_), 54.8 (d, ^2^*J*_PC_ = 7.4 Hz, CH_3_), 54.4 (d, ^2^*J*_PC_ = 7.6 Hz, CH_3_), 38.1 (CH_2_), 21.6 (CH_3_), 14.3 (CH_3_). ^31^P NMR (120 MHz, CDCl_3_): δ 21.9. ^19^F NMR (282 MHz, CDCl_3_) δ −63.4. FTIR (neat) ν_max_ 3261 (N-H), 1735 (C=O), 1327 (O=S=O), 1263 (P=O), 1163 (O=S=O). HRMS (ESI-TOF) *m/z*: calcd for C_21_H_26_F_3_NO_7_PS [M + H]^+^ 524.1114, found 524.1121.

*Ethyl 3-(dimethoxyphosphoryl)-3-((4-methylphenyl)sulfonamido)-3-(4-nitrophenyl)propanoate***(7q)**. The general procedure was followed, starting form imine **3q** (412 mg, 1 mmol) to afford 425 mg (85%) of **7q** as a white solid. M.p. (CH_2_Cl_2_-hexanes). 129–130 °C. ^1^H NMR (400 MHz, CDCl_3_) δ 7.96 (d, ^3^*J*_HH_ = 9.0 Hz, 2H), 7.56 (dd, ^3^*J*_HH_ = 9.0 Hz, ^3^*J*_PH_ = 2.3 Hz, 2H), 7.50 (d, ^3^*J*_HH_ = 8.2 Hz, 2H), 7.17 (d, ^3^*J*_HH_ = 8.2 Hz, 2H), 6.30 (d, ^3^*J*_PH_ = 10.3 Hz, 1H), 4.16 (m, 2H), 3.62 (dd, ^2^*J*_HH_ = 16.5 Hz, ^3^*J*_PH_ = 6.2 Hz, 1H), 3.58 (d, ^3^*J*_PH_ = 10.8 Hz, 3H), 3.55 (d, ^3^*J*_PH_ = 10.7 Hz, 3H), 3.48 (dd, ^2^*J*_HH_ = 16.5 Hz, ^3^*J*_PH_ = 8.2 Hz, 1H), 2.41 (s, 3H), 1.27 (t, ^3^*J*_HH_ = 7.2 Hz, 3H). ^13^C{^1^H} NMR (101 MHz, CDCl_3_) δ 169.7 (d, ^3^*J*_PC_ = 7.6 Hz, C_quat_), 147.4 (d, ^5^*J*_PC_ = 3.4 Hz, C_quat_), 144.1 (C_quat_), 142.5 (d, ^2^*J*_PC_ = 6.6 Hz, C_quat_), 138.8 (d, ^4^*J*_PC_ = 1.4 Hz, C_quat_), 129.4 (CH), 129.3 (d, ^3^*J*_PC_ = 5.0 Hz, CH), 127.5 (CH), 122.8 (d, ^4^*J*_PC_ = 2.6 Hz, CH), 62.3 (d, ^1^*J*_PC_ = 152.2 Hz, C_quat_), 61.4 (CH_2_), 54.9 (d, ^2^*J*_PC_ = 7.3 Hz, CH_3_), 54.6 (d, ^2^*J*_PC_ = 7.5 Hz, CH_3_), 38.3 (CH_2_), 21.7 (CH_3_), 14.2 (CH_3_). ^31^P NMR (120 MHz, CDCl_3_): δ 21.2. FTIR (neat) ν_max_ 3272 (N-H), 1743 (C=O), 1349 (O=S=O), 1244 (P=O), 1163 (O=S=O). HRMS (ESI-TOF) *m/z*: calcd for C_20_H_26_N_2_O_9_PS [M + H]^+^ 501.1091, found 501.1098.

*Ethyl 3-(5-chlorothiophen-2-yl)-3-(dimethoxyphosphoryl)-3-((4-methylphenyl)sulfonamido)propanoate***(7r)**. The general procedure was followed, starting form imine **3r** (407 mg, 1 mmol) to afford 378 mg (76%) of **7r** as a pale brown solid. M.p. (CH_2_Cl_2_-hexanes). 93–94 °C. ^1^H NMR (400 MHz, CDCl_3_) δ 7.48 (d, ^3^*J*_HH_ = 8.3 Hz, 2H), 7.17 (d, ^3^*J*_HH_ = 8.3 Hz, 2H), 6.79 (dd, ^3^*J*_HH_ = 4.0 Hz, ^4^*J*_PH_ = 3.4 Hz, 1H), 6.61 (d, ^3^*J*_HH_ = 4.0 Hz, 1H), 6.23 (d, ^3^*J*_PH_ = 7.2 Hz, 1H), 4.20 (q, ^3^*J*_HH_ = 7.2 Hz, 2H), 3.68 (d, ^3^*J*_PH_ = 10.7 Hz, 3H), 3.66 (d, ^3^*J*_PH_ = 10.5 Hz, 3H), 3.61 (m, 1H), 3.21 (dd, ^2^*J*_HH_ = 15.5 Hz, ^3^*J*_PH_ = 7.1 Hz, 1H), 2.40 (s, 3H), 1.30 (t, ^3^*J*_HH_ = 7.2 Hz, 3H). ^13^C{^1^H} NMR (101 MHz, CDCl_3_) δ 169.5 (d, ^3^*J*_PC_ = 5.6 Hz, C_quat_), 143.6 (C_quat_), 138.7 (d, ^4^*J*_PC_ = 1.4 Hz, C_quat_), 135.9 (d, ^2^*J*_PC_ = 7.7 Hz, C_quat_), 132.3 (d, ^5^*J*_PC_ = 3.6 Hz, C_quat_), 129.3 (CH), 128.6 (d, ^3^*J*_PC_ = 7.3 Hz, CH), 127.5 (CH), 125.3 (d, ^4^*J*_PC_ = 2.8 Hz, CH), 61.4 (CH_2_), 60.0 (d, ^1^*J*_PC_ = 161.5 Hz, C_quat_), 55.2 (d, ^3^*J*_PC_ = 7.3 Hz, CH_3_), 54.5 (d, ^3^*J*_PC_ = 7.6 Hz, CH_3_), 39.0 (CH_2_), 21.7 (CH_3_), 14.2 (CH_3_). ^31^P NMR (120 MHz, CDCl_3_): δ 20.3. FTIR (neat) ν_max_ 3270 (N-H), 1735 (C=O), 1341 (O=S=O), 1243 (P=O), 1163 (O=S=O). HRMS (ESI-TOF) *m/z*: calcd for C_18_H_24_ClNO_7_PS_2_ [M + H]^+^ 496.0415, found 496.0444..

*Ethyl 3-([1,1’-biphenyl]-4-yl)-3-(dimethoxyphosphoryl)-3-((4-methylphenyl)sulfonamido)propanoate***(7s)**. The general procedure was followed, starting form imine **3s** (443 mg, 1 mmol) to afford 478 mg (90%) of **7s** as a white solid. M.p. (CH_2_Cl_2_-hexanes). 112–113 °C. ^1^H NMR (400 MHz, CDCl_3_) δ 7.56–7.46 (m, 4H), 7.45–7.40 (m, 5H), 7.39–7.33 (m, 2H), 7.13 (d, ^3^*J*_HH_ = 8.3 Hz, 2H), 6.21 (d, ^3^*J*_PH_ = 10.5 Hz, 1H), 4.20 (q, ^3^*J*_HH_ = 7.2 Hz, 2H), 3.69 (dd, ^3^*J*_PH_ = 23.4 Hz, ^2^*J*_HH_ = 16.4 Hz, 1H), 3.56 (d, ^3^*J*_PH_ = 10.7 Hz, 3H), 3.49 (m, 1H), 3.46 (d, ^3^*J*_PH_ = 10.6 Hz, 3H), 2.38 (s, 3H), 1.29 (t, ^3^*J*_HH_ = 7.2 Hz, 3H). ^13^C{^1^H} NMR (101 MHz, CDCl_3_) δ 170.3 (d, ^3^*J*_PC_ = 7.3 Hz, C_quat_), 143.3 (C_quat_), 141.0 (d, ^5^*J*_PC_ = 3.1 Hz, C_quat_), 140.1 (d, ^4^*J*_PC_ = 1.4 Hz, C_quat_), 139.2 (C_quat_), 129.2 (CH), 129.0 (CH), 128.8 (d, ^3^*J*_PC_ = 5.1 Hz, CH), 128.3 (C_quat_), 127.8 (CH), 127.7 (CH), 127.1 (CH), 126.5 (d, ^4^*J*_PC_ = 2.8 Hz, CH), 62.1 (d, ^1^*J*_PC_ = 153.9 Hz, C_quat_), 61.1 (CH_2_), 54.7 (d, ^2^*J*_PC_ = 7.4 Hz, CH_3_), 54.2 (d, ^2^*J*_PC_ = 7.6 Hz, CH_3_), 38.2 (CH_2_), 21.7 (CH_3_), 14.3 (CH_3_). ^31^P NMR (120 MHz, CDCl_3_): δ 22.2. FTIR (neat) ν_max_ 3308 (N-H), 1729 (C=O), 1332 (O=S=O), 1241 (P=O), 1161 (O=S=O). HRMS (ESI-TOF) *m/z*: calcd for C_26_H_31_NO_7_PS [M + H]^+^ 532.1553, found 532.1555. 

*Methyl 3-(dimethoxyphosphoryl)-3-(3-fluorophenyl)-3-((4-methylphenyl)sulfonamido)propanoate***(12)**. The general procedure was followed, starting form imine **3k** (417 mg, 1 mmol) to afford 372 mg (81%) of **12** as a white solid. M.p. (CH_2_Cl_2_-hexanes). 115–116 °C. ^1^H NMR (400 MHz, CDCl_3_) δ 7.48 (d, ^3^*J*_HH_ = 8.3 Hz, 2H), 7.20–7.09 (m, 4H), 7.00 (m, 1H), 6.91 (m, 1H), 6.21 (br d, ^3^*J*_HH_ = 7.8 Hz, 1H), 3.73 (s, 3H), 3.64 (dd, ^3^*J*_PH_ = 24.4 Hz, ^2^*J*_HH_ = 16.2 Hz, 1H), 3.56–3.50 (m, 6H), 3.46 (dd, ^2^*J*_HH_ = 16.2 Hz, ^3^*J*_PH_ = 7.6 Hz, 1H), 2.39 (s, 3H). ^13^C{^1^H} NMR (101 MHz, CDCl_3_) δ 170.6 (d, ^3^*J*_PC_ = 6.8 Hz, C_quat_), 162.3 (dd, ^1^*J*_FC_ = 246.0 Hz, ^4^*J*_PC_ = 3.0 Hz, C_quat_), 143.7 (C_quat_), 139.0 (C_quat_), 137.1 (m, C_quat_), 129.5 (dd, ^3^*J*_FC_ = 8.1 Hz, ^4^*J*_PC_ 2.8 Hz, CH), 129.3 (CH), 127.5 (CH), 123.7 (dd, ^3^*J*_PC_ = 5.1 Hz, ^4^*J*_FC_ = 3.0 Hz, CH), 116.1 (dd, ^2^*J*_FC_ = 24.1 Hz, ^3^*J*_PC_ = 4.8 Hz, CH), 115.5 (dd, ^2^*J*_FC_ = 21.1 Hz, ^5^*J*_PC_ = 2.9 Hz, CH), 62.0 (dd, ^1^*J*_PC_ = 153.7 Hz, ^4^*J*_FC_ = 1.4 Hz, C_quat_), 54.9 (d, ^2^*J*_PC_ = 7.7 Hz, CH_3_), 54.4 (d, ^2^*J*_PC_ = 7.6 Hz, CH_3_), 52.3 (CH_3_), 38.2 (CH_2_), 21.6 (CH_3_). ^31^P NMR (120 MHz, CDCl_3_): δ 21.7. ^19^F NMR (282 MHz, CDCl_3_) δ −112.9. FTIR (neat) ν_max_ 3281 (N-H), 1729 (C=O), 1330 (O=S=O), 1243 (P=O), 1159 (O=S=O). HRMS (ESI-TOF) *m/z*: calcd for C_19_H_24_FNO_7_PS [M + H]^+^ 460.0990, found 460.1004. 

*Benzyl 3-(dimethoxyphosphoryl)-3-(3-fluorophenyl)-3-((4-methylphenyl)sulfonamido)propanoate***(13a)**. The general procedure was followed, starting form imine **3k** (417 mg, 1 mmol) to afford 449 mg (84%) of **13a** as a white solid. M.p. (CH_2_Cl_2_-hexanes). 78–79 °C. ^1^H NMR (400 MHz, CDCl_3_) δ 7.47 (d, ^3^*J*_HH_ = 8.2 Hz, 2H), 7.41–7.31 (m, 5H), 7.17–7.08 (m, 4H), 7.03–6.98 (m, 1H), 6.94–6.87 (m, 1H), 6.24 (d, ^3^*J*_PH_ = 10.2 Hz, 1H), 5.19 (d, ^2^*J*_HH_ = 12.3 Hz. 1H), 5.13 (d, ^2^*J*_HH_ = 12.3 Hz, 1H), 3.73–3.63 (m, 1H), 3.57–3.34 (m, 7H), 2.38 (s, 3H). ^13^C{^1^H} NMR (101 MHz, CDCl_3_) δ 169.9 (d, ^3^*J*_PC_ = 6.9 Hz, C_quat_), 162.3 (dd, ^1^*J*_FC_ = 246.0 Hz, ^4^*J*_PC_ = 3.0 Hz, C_quat_), 143.6 (C_quat_), 139.0 (d, ^4^*J*_PC_ = 1.3 Hz, C_quat_), 137.1 (C_quat_), 135.6 (C_quat_), 129.4 (dd, ^3^*J*_FC_ = 8.2 Hz, ^4^*J*_PC_ = 2.5 Hz, CH), 129.3 (CH), 128.7 (CH), 128.7 (CH), 128.5 (CH), 127.5 (CH), 123.7 (dd, ^3^*J*_PC_ = 5.0, ^4^*J*_FC_ = 2.9 Hz, CH), 116.0 (dd, ^2^*J*_FC_ = 24.0 Hz, ^3^*J*_PC_ = 4.7 Hz, CH), 115.4 (dd, ^2^*J*_FC_ = 21.1 Hz, ^5^*J*_PC_ = 3.1 Hz, CH), 67.0 (CH_2_), 61.9 (dd, ^1^*J*_PC_ = 153.6 Hz, ^4^*J*_FC_ = 1.6 Hz, C_quat_), 54.8 (d, ^2^*J*_PC_ = 7.5 Hz, CH_3_), 54.3 (d, ^2^*J*_PC_ = 7.7 Hz, CH_3_), 38.2 (CH_2_), 21.6 (CH_3_). ^31^P NMR (120 MHz, CDCl_3_): δ 21.6. ^19^F NMR (282 MHz, CDCl_3_) δ −112.9. FTIR (neat) ν_max_ 3281 (N-H), 1735 (C=O), 1331 (O=S=O), 1247 (P=O), 1154 (O=S=O). HRMS (ESI-TOF) *m/z*: calcd for C_25_H_28_FNO_7_PS [M + H]^+^ 536.1303, found 536.1322. 

*Benzyl 3-(dimethoxyphosphoryl)-3-((4-methylphenyl)sulfonamido)-3-phenylpropanoate***(13b)**. The general procedure was followed, starting form imine **3a** (367 mg, 1 mmol) to afford 476 mg (92%) of **13b** as a white solid. M.p. (CH_2_Cl_2_-hexanes). 84–85 °C. ^1^H NMR (400 MHz, CDCl_3_) δ 7.46 (d, *J* = 8.3 Hz, 2H), 7.37–7.29 (m, 7H), 7.20–7.15 (m, 1H), 7.12–7.06 (m, 4H), 6.22 (d, ^3^*J*_PH_ = 11.1 Hz, 1H), 5.14 (d, ^2^*J*_HH_ = 12.2 Hz, 1H), 5.09 (d, ^2^*J*_HH_ = 12.2 Hz, 1H), 3.65 (dd, ^3^*J*_PH_ = 22.3 Hz, ^2^*J*_HH_ = 16.6 Hz, 1H), 3.49 (dd, ^2^*J*_HH_ = 16.6 Hz, ^3^*J*_PH_ = 7.5 Hz, 1H), 3.37 (d, ^3^*J*_PH_ = 10.7 Hz, 3H), 3.31 (d, ^3^*J*_PH_ = 10.7 Hz, 3H), 2.34 (s, 3H). ^13^C{^1^H} NMR (101 MHz, CDCl_3_) δ 169.8 (d, ^3^*J*_PC_ = 8.1 Hz, C_quat_), 143.1 (C_quat_), 139.0 (d, ^4^*J*_PC_ = 1.4 Hz, C_quat_), 135.5 (C_quat_), 134.1 (d, ^2^*J*_PC_ = 7.3 Hz, C_quat_), 129.0 (CH), 128.4 (CH), 128.4 (CH), 128.2 (CH), 128.2 (CH), 128.0 (d, ^3^*J*_PC_ = 5.0 Hz, CH), 127.8 (d, ^4^*J*_PC_ = 2.6 Hz, CH), 127.4 (CH), 66.5 (CH_2_), 61.9 (d, ^1^*J*_PC_ = 154.1 Hz, C_quat_), 54.4 (d, ^2^*J*_PC_ = 7.4 Hz, CH_3_), 53.8 (d, ^2^*J*_PC_ = 7.7 Hz, CH_3_), 37.7 (CH_2_), 21.4 (CH_3_). ^31^P NMR (120 MHz, CDCl_3_): δ 22.0. FTIR (neat) ν_max_ 3322 (N-H), 1739 (C=O), 1337 (O=S=O), 1248 (P=O), 1163 (O=S=O). HRMS (ESI-TOF) *m/z*: calcd for C_25_H_29_NO_7_PS [M + H]^+^ 518.1397, found 518.1372. 

##### Procedure for the Obtention of 3-(Dimethoxyphosphoryl)-3-((4-methylphenyl)sulfonamido)-3-phenylpropanoic acid **14**

A mixture of aminophosphonate **13b** (518 mg, 1 mmol) and Pd-C 10% (106 mg, 0.1 mmol) in MeOH (50 mL) were stirred for 12 h under H_2_ atmosphere (75 psi). The mixture was then filtered on celite and concentrated under reduced pressure to yield product **14** as a white solid (402 mg, 94%), after crystallization in MeOH. M.p. (MeOH). 145–146 °C. ^1^H NMR (400 MHz, CDCl_3_) δ 9.91 (br s, 1H), 7.48 (d, ^3^J_HH_ = 8.2 Hz, 2H), 7.33 (d, *^3^J_HH_* = 7.6 Hz, 2H), 7.20 (m, 1H), 7.17–7.09 (m, 4H), 6.60 (d, *^3^J_PH_* = 10.6 Hz, 1H), 3.67 (dd, *^3^J_PH_* = 23.9 Hz, *^2^J_HH_* = 16.0 Hz, 1H), 3.52 (d, *^3^J_PH_* = 10.8 Hz, 3H), 3.49 (m, 1H), 3.47 (d, *^3^J_PH_* = 10.6 Hz, 3H), 2.38 (s, 3H). ^13^C{^1^H} NMR (101 MHz, CDCl_3_) δ 173.5 (d, *^3^J_PC_* = 7.7 Hz, C_quat_), 143.3 (C_quat_), 139.2 (d, *^4^J_PC_* = 1.4 Hz, C_quat_), 134.1 (d, *^2^J_PC_* = 7.6 Hz, C_quat_), 129.2 (CH), 128.4 (d, *^5^J_PC_* = 3.0 Hz, CH), 128.3 (d, *^3^J_PC_* = 5.1 Hz, CH), 128.0 (d, *^4^J_PC_* = 2.6 Hz, CH), 127.6 (CH), 62.1 (d, *^1^J_PC_* = 155.6 Hz, C_quat_), 55.1 (d, *^2^J_PC_* = 7.3 Hz, CH_3_), 54.5 (d, *^2^J_PC_* = 7.9 Hz, CH_3_), 38.0 (CH_2_), 21.6 (CH_3_). ^31^P NMR (120 MHz, CDCl_3_): δ 21.8. FTIR (neat) ν_max_ 3500–2500 (O-H st), 3271 (N-H st), 1714 (C=O st), 1337 (O=S=O st as), 1235 (P=O st), 1163 (O=S=O st sim) cm^–1^. HRMS (ESI-TOF) *m/z*: calcd for C_18_H_23_NO_7_PS [M + H]^+^ 428.0927, found 428.0901. 

##### Procedure for the Obtention of Dimethyl (1-((4-methylphenyl)sulfonamido)-1-phenylethyl)phosphonate **18**


A solution of **3a** (367 mg, 1 mmol) in dry CH_3_CN (3 mL) was stirred at room temperature under N_2_ atmosphere. To this mixture, Me_2_Zn (1.7 mL, 1.2 M in toluene, 2 mmol) was added and the mixture was stirred for 2h at room temperature. The reaction was quenched by a slow addition of a saturated aqueous solution of NH_4_Cl (1 mL) and dried over anhydrous MgSO_4_. The solid was removed by filtration and washed with AcOEt, and the filtrate was concentrated at reduced pressure to yield the crude product, which was purified by column chromatography in silica gel (AcOEt/Hexanes) to give 326 mg (85%) of **18** as a white solid [60]. M.p. (Et_2_O/pentane). 163–164 °C. Lit. 161–162 (Et_2_O). ^1^H NMR (400 MHz, CDCl_3_) δ 7.53 (d, *^3^J_HH_* = 8.3 Hz, 2H), 7.40 (m, 2H), 7.22–7.18 (m, 3H), 7.11 (d, *^3^J_HH_* = 8.2 Hz, 2H), 5.78 (d, *^3^J_PH_* = 8.1 Hz, 1H), 3.70 (d, *^3^J_PH_* = 10.4 Hz, 3H), 3.35 (d, *^3^J_PH_* = 10.4 Hz, 3H), 2.36 (s, 3H), 1.97 (d, ^3^J_PH_ = 16.8 Hz, 3H). ^13^C {^1^H} NMR (75 MHz, CDCl_3_) δ 142.3 (C_quat_), 141.8 (d, *^4^J_PC_* = 1.7 Hz, C_quat_), 133.4 (C_quat_), 128.6 (CH), 128.4 (d, *^3^J_PC_* = 6.0 Hz, CH), 128.1 (d, *^4^J_PC_* = 2.2 Hz, CH), 128.0 (d, *^5^J_PC_* = 2.9 Hz, CH), 126.7 (CH), 61.1 (d, *^1^J_PC_* = 152.2 Hz, C_quat_), 54.5 (d, *^2^J_PC_* = 7.1 Hz, CH_3_), 53.9 (d, *^2^J_PC_* = 7.0 Hz, CH_3_), 21.3 (CH_3_), 20.4 (d, *^2^J_PC_* = 5.2 Hz CH_3_). ^31^P NMR (120 MHz, CDCl_3_) δ 26.1. FTIR (neat) ν_max_ 3315 (N-H), 1327 (O=S=O), 1242 (P=O), 1166 (O=S=O). HRMS (ESI-TOF) *m/z*: calcd. for C_17_H_22_NO_5_PS [M + Na]^+^ 406.0848, found 406.0856. 

### 3.2. Biology

#### 3.2.1. Materials

Reagents and solvents were used as purchased without further purification. All stock solutions of the investigated compounds were prepared by dissolving the powered materials in appropriate amounts of dimethylsulfoxide (DMSO). The final concentration of DMSO never exceeded 5% (*v*/*v*) in the reactions. The stock solution was stored at 5 °C until it was used.

#### 3.2.2. Cell Culture

Human epithelial lung carcinoma cells (A549) (ATCC^®^ CCL-185™, ATCC, Manassas, VA, USA) were grown in Kaighn’s Modification of Ham’s F-12 Medium (ATCC^®^ 30-2004™, ATCC, Manassas, VA, USA) and lung fibroblast cells (MRC5) (ATCC^®^ CCL-171™, ATCC, Manassas, VA, USA) were grown in Eagle’s Minimum Essential Medium (EMEM, ATCC^®^ 30-2003™, ATCC, Manassas, VA, USA). Epithelial ovary adenocarcinoma cells (SKOV3) (ATCC^®^ HTB-77™, ATCC, Manassas, VA, USA) were grown in McCoy’s 5A medium (ATCC^®^ 30-2007™, ATCC, Manassas, VA, USA). All of them were supplemented with 10% of fetal bovine serum (FBS) (Sigma-Aldrich, Madrid, Spain) and with 1% of NORMOCIN solution (Thermo Fisher, Waltham, MA, USA). Cells were incubated at 37 °C and 5% CO_2_ atmosphere, and were split every 3–4 days to maintain monolayer coverage. For the cytotoxicity experiments, the A549 and SKOV3 cells were seeded in 96-well plates at a density of 2.5–3 × 10^3^ cells per well and incubated overnight to achieve 70% of confluence at the time of exposition to the cytotoxic compound.

#### 3.2.3. Cytotoxicity Assays

Cells were exposed to different concentrations of the cytotoxic compounds and were incubated for 48 h. Then, 10 µL of cell counting kit-8 was added into each well for an additional two hours’ incubation at 37 °C. The absorbance of each well was determined by an Automatic Elisa Reader System (Thermo Scientific Multiskan FC Automatic Elisa Reader System, Thermo Scientific, Shanghai, China) at 450 nm wavelength.

## 4. Conclusions

In conclusion, we report an efficient methodology for the preparation of phosphonate analogs of aspartic acid, holding a variety of substituents at their α-aromatic ring. α-Ketiminophosphonates are generated by the oxidation of their parent tertiary α-aminophosphonates and a subsequent aza-Reformatsky reaction with alkyl iodoacetate derivatives. Moreover, this methodology has been successfully extended to aldimines and activated ketimines, affording the Reformatsky products in high yields. This strategy allows the possibility of assorted structural diversity in the resultant scaffold depending on the starting imine or alkyl iodoacetate. Moreover, the phosphorated analogues of aspartic acid **6** showed in vitro cytotoxicity inhibiting the growth of human tumor cell lines SKOV3 (human ovarian carcinoma) and A549 (carcinomic human alveolar basal epithelial cell), and a high selectivity toward the MRC5 non-malignant lung fibroblasts. The most active substrates were proved to be ethyl ester derivatives. Although *p*-tolyl derivatives showed the best result against A549, the introduction of a trifluoromethylphenyl moiety in the para position exhibited the most remarkable IC_50_ value against the SKOV3 cell line. Moreover, the majority of the compounds were selective toward the non-malignant cells. The best IC_50_ values obtained are 9.80 µM in the SKOV3 cell line for α-aminophosphonate **7p**, with a *p*-trifluorophenyl substituent, and 0.34 µM in the A549 cell line for substrate **7k**, holding a *p*-methylphenyl moiety. Most of the compounds presented in this study show low micromolar activity and a high selectivity toward the non-malignant cells. It has also been proved that the absolute configuration of the tetrasubstituted stereocenter does not have any influence on the biological activity of the phosphorated aspartic acid derivatives, since both enantiomers of substrate **7k** showed similar IC_50_ values.

## Data Availability

The data presented in this study are available in the Appendix A or on request from the corresponding author (^1^H, ^13^C, ^19^F and ^31^P-NMR and HRMS spectra and cytotoxicity essays).

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
