# Peer review of "Synthesis of Tetrasubstituted Phosphorus Analogs of Aspartic Acid as Antiproliferative Agents"

_molecules, 2022, doi:10.3390/molecules27228024_

Round 1
Reviewer 1 Report
J. Vicario and co-workers reported Synthesis of Tetrasubstituted Phosphorus Analogs of Aspartic Acid as Antiproliferative Agents. The synthesis of these compounds has already reported by the author in Org. Lett. 2019, 21, 9473−9477(author mentioned as ref 51). In this report just they are reporting the biological activity of these products. therefore in my opinion this report may not be fit for this journal, it's better to publish in only biology related journals.
Author Response
We thank the reviewer for the suggestion of publishing in “only biology related journals”. However, we would like to make an effort to change referee Nr 1’s mind in the convenience of publishing our paper in Molecules (please consider our comments from a friendly point of view):
As stated in the “aims and scope” of Molecules, the main research areas include (but are not limited to): Organic chemistry and Medicinal chemistry, among others. For this reason, we think that our research on the synthesis and biological evaluation of tetrasubstituted phosphorus analogs of aspartic acid fits with the scope of the journal.
In addition, considering that the special issue projected for the publication of this research is named “Organophosphorus Chemistry: A New Perspective”, we consider our manuscript specially appropriated to be published in this issue in Molecules. In fact, we demonstrate that the presence of a phosphorus moiety is superior to a carboxylate group in terms of activity, which calls attention to the importance of organophosphorus compounds, which we assume is the intention of this special issue.
Moreover, in this article the previously reported methodology has been extended to other substrates, including not only other imines but also other iodoacetate derivatives (Schemes 2 and 5). In addition, during the revision process, we have performed additional experiments including control reactions for a further understanding of the reaction pathway. Therefore, the revised version includes not only biological experiments but also new synthetic applications of our previous report in Org. Lett.
Reviewer 2 Report
The manuscript entitled “Synthesis of Tetrasubstituted Phosphorus Analogs of Aspartic Acid as Antiproliferative Agents” by Xabier Del Corte et al., describes the synthesis of aspartic acid analogs combined with α-aminophosphonate fragment via the aza-Reformatsky reaction. The synthesized aspartic acid analogs were tested in vitro as effective antiproliferative agents on two cancer cell lines. The manuscript is well-written and has a clear structure.
Overall, the manuscript cannot be accepted for publication in the current form and requires major revision. In addition, I have the following comments and suggestions for the authors:
1. Chemistry. The presented synthetic pathway is not new, and was published in ‘Org. Lett. 2019, 21, 23, 9473–9477’. Also, there is a question why Authors did not use enantioselective catalysts in this manuscript?
2. Authors should carefully check 1H and 13C NMR spectra data for missing signals.
3. Biology. The presented results are not significant compared to Doxorubicin.
4. Authors should combine tables 1-3 in one table. Also, check the IC50 value for Doxorubicin.
Author Response
We thank the reviewer for the comments. In the following lines, we will answer the comments one by one.
- The presented synthetic pathway is not new, and was published in ‘Org. Lett. 2019, 21, 23, 9473–9477’. Also, there is a question why Authors did not use enantioselective catalysts in this manuscript?
We agree with referee Nr. 2 that the synthesis of all substrates could have been done in an enantioselective fashion for all compounds. However, consider the waste of enantiopure catalysts for the preparation of all compounds in an enantioenriched form. For this reason, first, we decided to evaluate the racemic compounds in order to select the most active ones. In Scheme 7 we illustrate the application of the enantioselective methodology to the synthesis of both enantiomers of one of the described molecules, along with the IC50 of both isomers, which showed to have equal cytotoxic activity. Herein we report a general protocol for the synthesis of phosphorated analogs of aspartic acid using a cheap racemic phosphoric acid derivative. The value of the research should be seen as a general, efficient and cheap, protocol for the access to racemic analogs of aspartic acid and the discovery of a new family of potential cytotoxic agents.
Authors should carefully check 1H and 13C NMR spectra data for missing signals.
We have double checked the NMR spectra.
- The presented results are not significant compared to Doxorubicin.
Doxorubicin is a chemotherapeutic agent approved for the treatment of several cancer diseases. Its IC50 value is usually used as reference in cytotoxic activity essays. However, the fact that our compounds do not beat doxorubicin does not mean that the cytotoxicity of our compounds is not significant. It would have been great to discover a new family of compounds with higher cytotoxicity than an approved drug. However the fact that the anticancer activity of a new structure is comparable (although lower, we agree) to doxorubicin is itself remarkable and may be an starting point in the development of new chemotherapeutic agents.
- Authors should combine tables 1-3 in one table. Also, check the IC50value for Doxorubicin.
IC50 value for Doxorubicin has been corrected. We have combined tables 2 and 3 in one.
Reviewer 3 Report
The authors have described the synthesis of tetrasubstituted phosphorus analogs of aspartic acid as antiproliferative agents. They have presented the manuscript well, and the current version is suitable for molecules. Below are some minor concerns regarding this manuscript.
|
1. |
The authors published similar kind of work in Organic Letters 2019 21 (23), 9473-9477. The authors only just studied the anticancer activity, so how the authors could explain the novelty of the current work. |
|
2. |
Abstract: Report which compound showed the potential activity. |
|
3. |
Schemes: I have not seen the reaction timings in the Schemes; please include them. |
|
4. |
How stable are these compounds in the column chromatography? |
|
5. |
What are the most prominent structural features of activity? |
Author Response
We thank the reviewer for the valuable comments. In the following lines we will answer the comments one by one.
- The authors published similar kind of work in Organic Letters 2019 21 (23), 9473-9477. The authors only just studied the anticancer activity, so how the authors could explain the novelty of the current work.
We agree that an enantioselective reformatsky reaction with ketiminophosphonates has been already reported. However, herein we report a general protocol for the synthesis of phosphorated analogs of aspartic acid using a cheap BINOL. The value of the research should be seen as a general, efficient and cheap, protocol for the access to racemic analogs of aspartic acid. In addition, in this case, we have studied the reaction in deep and have explored the extension of this methodology to several aldimines and ketiminoesters (Scheme 2), and to some iodoacetate derivatives (Scheme 5). In the revision, we have added some unsuccessful tests to the manuscript, in order to understand the nature of the reaction.
- Abstract: Report which compound showed the potential activity.
The compound which showed the potential activity and its IC50 value has been added in the abstract.
- Schemes: I have not seen the reaction timings in the Schemes; please include them.
The reaction times have been added in all the schemes.
- How stable are these compounds in the column chromatography?
All the presented compounds have been purified by column chromatography on silica gel and none of them showed degradation or decomposition during the process.
- What are the most prominent structural features of activity?
A sentence has been added in the conclusions: “The most active subtrates were proved to be ethyl ester derivatives. Although, p-tolyl derivative showed the best result against A549, the introduction of a trifluoromethylphenyl moiety in para position exhibited the most remarkable IC50 value against SKOV3 cell line. Moreover, the majority of the compounds were selective towards non malignan cells.”
Round 2
Reviewer 1 Report
After major revision author made adequate responses both experimentally and in manuscript preparation. Therefore, I am convinced to accept the manuscript in its current form.
Author Response
We thank the reviewer for the positive comments.
Reviewer 2 Report
The manuscript entitled “Synthesis of Tetrasubstituted Phosphorus Analogs of Aspartic Acid as Antiproliferative Agents” by Xabier Del Corte et al. Version # 2.
Comments and suggestions for Authors. The manuscript has been submitted to an Organic Chemistry Section of the Molecules journal. The synthetic part of the manuscript was published previously. Therefore, there is no novelty for organic chemistry. I recommend deepening biological research, and re-submit improved manuscript to the Medicinal Chemistry section.
Author Response
We thank the reviewer for the comments.